# Holistic Physics Solver: Learning PDEs in a Unified Spectral-Physical Space

**Xihang Yue** [1] [2]   **Yi Yang** [1] [2]   **Linchao Zhu** [1] [2]

## Abstract

Recent advances in operator learning have produced two distinct approaches for solving partial differential equations (PDEs): attention-based methods offering point-level adaptability but lacking spectral constraints, and spectral-based methods providing domain-level continuity priors but limited in local flexibility. This dichotomy has hindered the development of PDE solvers with both strong flexibility and generalization capability. This work introduces Holistic Physics Mixer (HPM), a simple framework that bridges this gap by integrating spectral and physical information in a unified space. HPM unifies both approaches as special cases while enabling more powerful spectral-physical interactions beyond either method alone. This enables HPM to inherit both the strong generalization of spectral methods and the flexibility of attention mechanisms while avoiding their respective limitations. Through extensive experiments across diverse PDE problems, we demonstrate that HPM consistently outperforms state-of-the-art methods in both accuracy and computational efficiency, while maintaining strong generalization capabilities with limited training data and excellent zero-shot performance on unseen resolutions.

## 1. Introduction

Solving partial differential equations (PDEs) efficiently remains a major challenge in scientific computing and engineering applications. Traditional numerical solvers rely on high-precision meshes and substantial computational resources, making them impractical for many real-world scenarios. To address this challenge, neural operators (Lu et al., 2019; Li et al., 2020; Tripura & Chakraborty, 2022) have

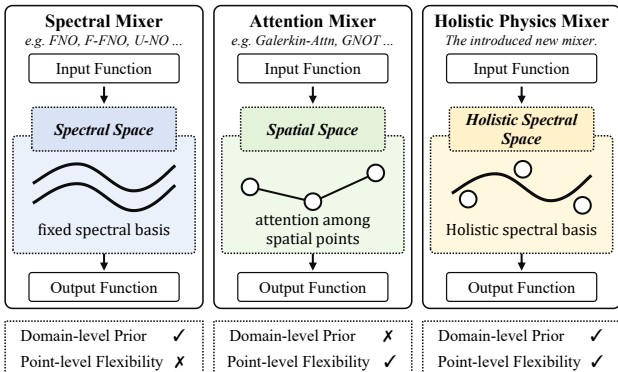

**Different Neural Modules for Learning Operator Mappings**

*Figure 1.* HPM combines domain-level spectral structure and point-wise physical states in a unified holistic spectral space, enabling more powerful feature learning beyond either approach alone.

emerged as a data-driven approach that learns continuous mappings between function spaces for solving parametric PDEs.

Recent works have developed two main categories of neural operators. **Spectral-based methods** (Li et al., 2020; Kovachki et al., 2023; Tran et al., 2021) can efficiently learn operator mappings with limited training data by approximating physical functions in truncated spectral spaces. However, their fixed spectral processing mechanism lacks point-level flexibility (ability to process individual spatial points differently based on local features), making them less effective at capturing high-frequency details and fine-scale variations in physical systems (George et al., 2024; Qin et al., 2024). In contrast, **attention-based methods** (Hao et al., 2023; Xiao et al., 2023; Wu et al., 2024) can flexibly learn various physical dynamics through point-wise learning in physical domains, achieving superior performance with sufficient training data. However, their pure data-driven framework without spectral priors tends to overfit with scarce training data and has limited generalization capability (Xiao et al., 2023), leading to significant performance degradation.

This work introduces Holistic Physics Mixer (HPM) [1], a framework that unifies spectral and physical information in a single holistic space. The key contribution lies in our holistic spectral space design that simultaneously encodes

[1]The State Key Lab of Brain-Machine Intelligence, Zhejiang University, Hangzhou, China [2]College of Computer Science and Technology, Zhejiang University, Hangzhou, China. Correspondence to: Linchao Zhu <zhulinchao@zju.edu.cn>.

*Proceedings of the 42[st] International Conference on Machine Learning*, Vancouver, Canada. PMLR 267, 2025. Copyright 2025 by the author(s).

---

[1]Code: https://github.com/yuexihang/HPM

both domain-level spectral structure and point-wise physical states. This strategy naturally balances global continuity constraints with local adaptivity. Through a learnable coupling mechanism, HPM allows each spatial location to adaptively modulate different spectral components while maintaining the benefits of spectral learning. This unified treatment enables HPM to inherit both the strong generalization of spectral methods and the flexibility of attention mechanisms while avoiding their respective limitations.

Through extensive experiments across diverse PDE problems, we demonstrate that HPM: (a) consistently outperforms both spectral-based and attention-based methods, (b) maintains strong generalization with limited training data and excellent resolution generalization capability like spectral methods, (c) efficiently utilizes increased data like attention methods. (d) Our visualization analysis shows that HPM learns meaningful spectral modulation patterns that could guide the design of fixed spectral neural operators.

Our core contributions are summarized as follows:

- We present Holistic Physics Mixer (HPM), a unified framework that processes features in holistic spectral space, integrating both domain-level structure and point-wise physical states.

- We present an effective implementation of HPM that (a) maintains strong performance with limited training data and excellent resolution generalization through spectral priors, and (b) efficiently handles increased training data and fine-scale status variations in physical space via integrating point-wise adaptivity.

- We validate HPM's superiority in various scenarios through comprehensive experiments, and find the spectral processing patterns learned by HPM can help design fixed spectral neural operators.

## 2. Background and Related Work

### 2.1. Neural Operator Learning

In neural operator learning (Lu et al., 2021; Li et al., 2020; Kovachki et al., 2023), the solution of parametric partial differential equations is formulated as the operator mapping between two infinite-dimensional function spaces:

$$\mathcal{G}^{\dagger} : \mathcal{A} \to \mathcal{U}, \tag{1}$$

$$\mathcal{A} = \{\boldsymbol{a} | \boldsymbol{a} : \Omega \to \mathbb{R}^{d_a}\}, \mathcal{U} = \{\boldsymbol{u} | \boldsymbol{u} : \Omega \to \mathbb{R}^{d_u}\}, \tag{2}$$

where $\Omega$ denotes the physical domain, $d_a$ and $d_u$ represent the channel number of input functions and output functions respectively. For example, in the steady-state problem Darcy Flow, $\boldsymbol{a}$ denotes the diffusion coefficient and $\boldsymbol{u}$ represents the solution function. In the time-series problem Navier-Stokes, $\boldsymbol{a}$ is the vorticity states in previous time steps and $\boldsymbol{u}$ is the vorticity states of following time steps.

The operator learning problem aims to learn a parameterized surrogate model $\mathcal{G}_{\boldsymbol{\theta}}^{\dagger}$ for approximating the operator mapping $\mathcal{G}^{\dagger}$. As established in previous works (Li et al., 2020; Wu et al., 2024), the neural operator is commonly implemented as a stack of network layers:

$$\mathcal{G}_{\boldsymbol{\theta}}^{\dagger} = V \circ M_l \circ ... \circ M_2 \circ M_1 \circ P, \tag{3}$$

where $P$ and $V$ are element-wise projecting layers that map between the input/output functions and latent functions. $M_i$ takes $\boldsymbol{v}_{i-1} \in \mathbb{R}^{N \times d_v}$ as input and produces $\boldsymbol{v}_i \in \mathbb{R}^{N \times d_v}$ as output, mixing features along both token and channel dimensions. In previous works (Tran et al., 2021; Wu et al., 2024), $M_i$ is formulated as:

$$\boldsymbol{v}_{i-1}^{\text{mid}} = \mathcal{F}^{\text{mixer}}(\text{LayerNorm}(\boldsymbol{v}_{i-1})) + \boldsymbol{v}_{i-1}, \tag{4}$$

$$\boldsymbol{v}_i = \text{FeedForward}(\text{LayerNorm}(\boldsymbol{v}_{i-1}^{\text{mid}})) + \boldsymbol{v}_{i-1}^{\text{mid}}, \tag{5}$$

where $\mathcal{F}^{\text{mixer}}$ represents operations like spectral processing or self-attention for mixing spatial information. Various neural mixers (Li et al., 2020; Wu et al., 2024; Raonic et al., 2024) have been explored for neural operator learning.

### 2.2. Spectral-based Neural Operators

Unlike the classical neural modules such as CNN (LeCun et al., 1995), RNN (Chung et al., 2014), and Self-Attention mechanism (Vaswani, 2017) that directly mix features of spatial tokens in the spatial domain, a significant line of works (Li et al., 2020; Lee-Thorp et al., 2021; Guibas et al., 2021; Yue et al., 2024; Liu et al., 2024b; Gao et al., 2025) explores learning operator mappings in spectrum space, which reduces learning difficulty through efficient function approximation with spectral basis functions. The spectral-based token mixer can be formulated as:

$$\mathcal{F}_{\text{spectral}}^{\text{mixer}}(\boldsymbol{x}) = \mathcal{T}_{\text{fourier}}^{-1} \circ \text{Project} \circ \mathcal{T}_{\text{fourier}}(\boldsymbol{x}), \tag{6}$$

where $\mathcal{T}_{\text{fourier}}(\cdot)$ represents the spectral transform operator such as Fourier Transform, yielding spectral feature $\hat{\boldsymbol{x}} \in \mathbb{R}^{k \times d_v}$. $k$ represents the number of retained frequencies in the spectral domain. $\mathcal{T}_{\text{fourier}}^{-1}(\cdot)$ denotes the inverse spectral transform operator. Project represents operations in the spectral domain, commonly including simple fully connected layers and normalization operations.

Different spectral transforms have been explored: FNO (Li et al., 2020) learns operators in Fourier spectral space, LNO (Cao et al., 2024) learns in Laplacian spectral space, and WMT (Gupta et al., 2021) learns in wavelet spectral space. Additional works investigate complex physical domain processing (Li et al., 2023a; Bonev et al., 2023; Lingsch et al., 2023; Liu et al., 2024a;c; 2023a), computational efficiency enhancement (Poli et al., 2022; Tran et al., 2021; Wang & Wang, 2024), and multi-scale feature processing (Rahman et al., 2022; Zhang et al., 2024).

However, spectral-based methods employ static spectral eigenfunctions, which restricts their point-level adaptability and makes them struggle to process high-frequency details and fine-scale status variations in physical systems. As training data increases, the fixed frequency design limits significant improvement in prediction accuracy.

## 2.3. Attention-based Neural Operators

Recent works extensively explore learning operator mappings based on attention mechanisms (Vaswani, 2017), enabling flexible handling of diverse physical domains. Given the input feature $x \in \mathbb{R}^{N \times d_v}$, the vanilla attention mechanism could be simply formulated as follows:

$$\mathcal{F}_{\text{attn}}^{\text{mixer}} = \text{Norm}(\mathrm{Q}(x)\mathrm{K}^T(x))\mathrm{V}(x), \quad (7)$$

where $\mathrm{Q}(\cdot)$, $\mathrm{K}(\cdot)$ and $\mathrm{V}(\cdot)$ are all fully connected layers that map the feature $x$ into $x^q \in \mathbb{R}^{N \times d_{qk}^{\text{attn}}}$, $x^k \in \mathbb{R}^{N \times d_{qk}^{\text{attn}}}$ and $x^v \in \mathbb{R}^{N \times d_v^{\text{attn}}}$ respectively. $\text{Norm}(\cdot)$ represents the normalization operation such as Softmax.

To address the quadratic complexity of attention, previous works (Li et al., 2023b; Cao, 2021; Hao et al., 2023; Calvello et al., 2024) employ efficient attention variants. Factformer (Li et al., 2023c) enhances efficiency with multi-dimensional factorized attention. Additionally, HT-Net (Liu et al., 2023b) introduces hierarchical transformers for better multi-scale features, and ONO (Xiao et al., 2023) improves generalization with orthogonal attention. And some works (McCabe et al., 2023; Ye et al., 2024; Hao et al., 2024; Chen et al., 2024; Li et al., 2024; Alkin et al., 2024; Shen et al.) have explored large scale pretraining for neural operators with attention-based methods.

While attention-based methods achieve impressive performance on various PDEs (Wu et al., 2024), their lack of spectral constraints results in subpar performance under limited training data and unseen resolution samples compared to spectral-based methods. This motivates our work to combine the advantages of both approaches.

## 3. Methodology

We introduce Holistic Physics Mixer, a unified form of spectral-based and attention-based neural operators, for integrating complementary advantages of both approaches.

### 3.1. Holistic Physics Mixer: A Unified Formulation

Physical systems often involve complex field quantities represented as features $x \in \mathbb{R}^{N \times d}$ defined on a discrete sampling of the physical domain $\Omega$ with $N$ points, where each point is characterized by a $d$-dimensional state vector encoding physical properties. Traditional approaches to processing such features often rely on fixed spectral transforms (e.g., FNO) that fail to capture the complex interplay between local physical states and global domain structure.

**Holistic Spectral Space.** The key insight is that effective feature processing should simultaneously consider both point-wise physical states and domain-level spectral properties. We achieve this through a new coupling function $\mathbf{H} : \mathbb{R}^{N \times d} \times \mathbb{R}^{N \times k} \to \mathbb{R}^{N \times k}$ (where $k$ is the number of spectral basis functions) that integrates physical states $x$ with spectral basis functions $\mathbf{\Phi} \in \mathbb{R}^{N \times k}$ (e.g., Fourier basis) to generate adaptive spectral eigenfunctions. This coupling enables a unified representation space where features dynamically respond to local physical variations while preserving global spectral properties.

**Holistic Physics Mixer.** Based on holistic spectral space, we propose the Holistic Physics Mixer $\mathcal{F}_{\text{Holi}}^{\text{mixer}}$ that adaptively processes features through spectral transformation:

$$\mathcal{F}_{\text{Holi}}^{\text{mixer}}(x) = \mathcal{T}_{\text{HPT}}^{-1} \circ \text{Project} \circ \mathcal{T}_{\text{HPT}}(x), \quad (8)$$

$$\mathcal{T}_{\text{HPT}}(x) = \mathbf{H}(x, \mathbf{\Phi})^T x, \quad (9)$$

$$\mathcal{T}_{\text{HPT}}^{-1}(\hat{x}) = \mathbf{H}(x, \mathbf{\Phi})\hat{x}. \quad (10)$$

Here, $\mathcal{T}_{\text{HPT}}$ and $\mathcal{T}_{\text{HPT}}^{-1}$ denote our proposed Holistic Physics Transform and its inverse operation. The transform maps input features to a transformed representation $\hat{x} \in \mathbb{R}^{h \times d}$ in the holistic spectral space. This dual encoding in $\hat{x}$ - capturing both intrinsic domain structure and point-specific variations - establishes a new paradigm for operator learning that naturally balances global continuity constraints with local adaptivity. Additionally, Tang et al. (2024) shows that integrating spectral and spatial features potentially improves the model's robustness.

**Degraded Cases.** Both spectral and attention based methods can be viewed as simplified cases of Holistic Physics Mixer $\mathcal{F}_{\text{Holi}}^{\text{mixer}}$ by degraded instantiations of $\mathbf{H}$:

- Fixed Spectral Neural Operators. When $\mathbf{H}(x, \mathbf{\Phi}) = \mathbf{\Phi}$, $\mathcal{F}_{\text{Holi}}^{\text{mixer}}$ reduces to fixed spectral neural operators like FNOs (Li et al., 2020; Kovachki et al., 2023). This classical formulation enforces strong domain-level structure but capability for capturing fine-scale variations in physical systems and utilizing enough training samples.

- Linear Attention Mechanism. When $\mathbf{H}(x, \mathbf{\Phi}) = \psi(\text{MLP}(x))$ where $\psi$ is a normalization function, $\mathcal{F}_{\text{Holi}}^{\text{mixer}}$ becomes equivalent to linear attention mechanism (Katharopoulos et al., 2020; Cao, 2021). While this enables flexible point-wise processing, it fails to leverage domain-level continuity constraints, potentially compromising stability and generalization.

These special cases reveal how existing approaches suboptimally prioritize either domain structure $\mathbf{\Phi}$ or point states $x$. To overcome these limitations, we introduce a few instantiations of coupling functions $\mathbf{H}$ that effectively harmonize

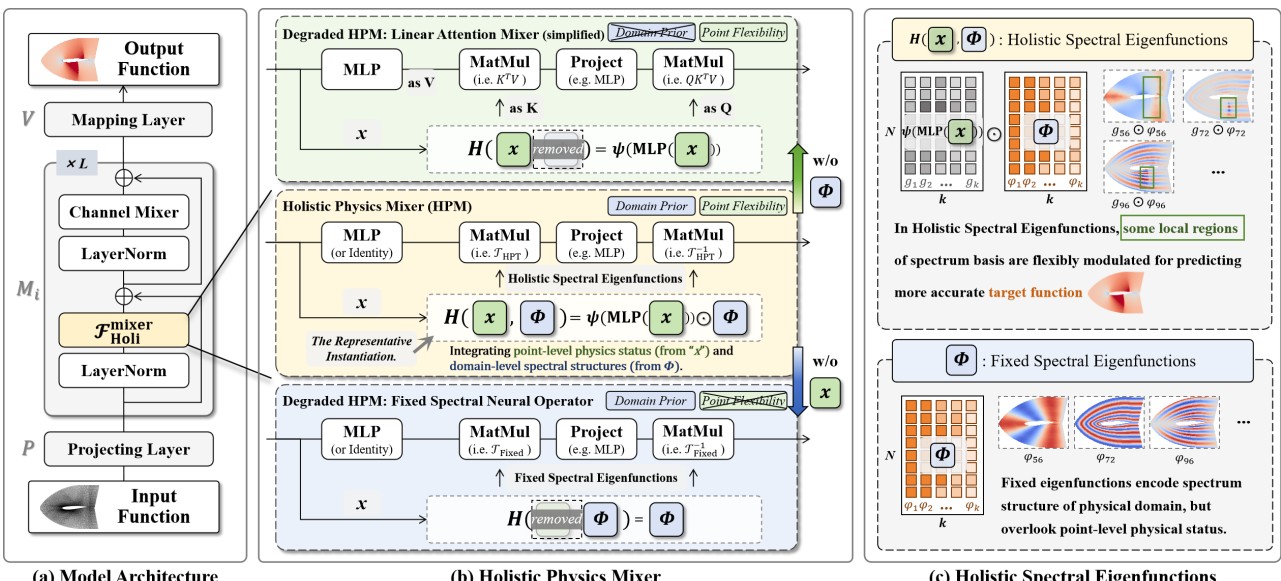

*Figure 2.* (a) Overall architecture of Holistic Physics Solver. (b) Structure of Holistic Physics Mixer. (c) Visualization of Fixed Spectral Eigenfunctions and Holistic Spectral Eigenfunctions.

spectral structure with point-wise adaptivity.

**Difference with Spectral Modes Selection.** While some previous works (Guibas et al., 2021; George et al., 2024) also incorporate data-dependent modulation on spectral features, HPM differs with them in several aspects: (a) Different Objective. Guibas et al. (2021); George et al. (2024) aim to improve *efficiency and robustness* of spectral features by sparsifying the frequency modes. In contrast, HPM focuses on *enhancing the flexibility of preset spectral features* via point-wise modulation. (b) Different Methodology. Guibas et al. (2021); George et al. (2024) select modes in the spectral domain post-transformation, while HPM introduces spatial domain modulation pre-transformation, enabling point-wise flexibility while preserving spectral structure. (c) Different Application. AFNO (Guibas et al., 2021) targets computer vision tasks and iFNO (George et al., 2024) focuses large-scale meshes, whereas HPM addresses the PDE learning challenge requiring both global spectral coherence and local adaptivity.

These spectral processing techniques (Guibas et al., 2021; George et al., 2024) could complement HPM, potentially combining both benefits for diverse applications.

### 3.2. Instantiations

The coupling function $\mathbf{H}(\boldsymbol{x}, \boldsymbol{\Phi})$ in Holistic Physics Mixer can be implemented in various ways to achieve different balances between spectral structure and point-wise flexibility. In this section, we explored several designs to find the effective way to integrate spectral structure with point-wise adaptivity under minimal overhead:

**Form 1: Point-wise Softmax Coupling.** A straightforward approach using point-wise coupling with Softmax normalization:

$$\mathbf{H}(\boldsymbol{x}, \boldsymbol{\Phi}) = \text{Softmax}(\text{MLP}(\boldsymbol{x})) \odot \boldsymbol{\Phi}, \qquad (11)$$

where $\odot$ represents element-wise multiplication. The Softmax ensures balanced feature scales and lets each point adaptively focus on different spectral components. This design provides an effective balance of stability and flexibility, making it our preferred choice.

**Form 2: Point-wise Sigmoid Coupling.** A variation using Sigmoid instead of Softmax:

$$\mathbf{H}(\boldsymbol{x}, \boldsymbol{\Phi}) = \text{Sigmoid}(\text{MLP}(\boldsymbol{x})) \odot \boldsymbol{\Phi}, \qquad (12)$$

This removes the sum-to-1 constraint of Softmax, allowing multiple frequency components to be active simultaneously.

**Form 3: Global-pooled Coupling.** An approach that first aggregates spatial information:

$$\mathbf{H}(\boldsymbol{x}, \boldsymbol{\Phi}) = \text{Softmax}(\text{AvgPool}(\text{MLP}(\boldsymbol{x}))) \odot \boldsymbol{\Phi}, \qquad (13)$$

While computationally efficient, this global pooling sacrifices the fine-grained adaptivity that makes point-wise coupling effective.

**Form 4: Additive Coupling.** A simple additive approach:

$$\mathbf{H}(\boldsymbol{x}, \boldsymbol{\Phi}) = \text{Softmax}(\text{MLP}(\boldsymbol{x})) + \boldsymbol{\Phi}, \qquad (14)$$

This directly adds transformed point-wise features to the spectral basis.

**Form 5: Concatenation Coupling.** A direct concatenation approach:

$$\mathbf{H}(\boldsymbol{x}, \boldsymbol{\Phi}) = \text{Softmax}(\text{MLP}(\text{Concat}(\boldsymbol{x}, \boldsymbol{\Phi}))), \quad (15)$$

This processes the physical states and spectral basis jointly by concatenating them before applying the transformation.

**Discussion.** Through empirical evaluation (Table 5), we found that the first form - Point-wise Softmax Coupling - achieves superior performance. Although simple, this design effectively balances several key requirements: (a) The point-wise coupling enables adaptive spectral processing at each spatial location. (b) The Softmax normalization ensures stable training and consistent feature scales. (c) The direct multiplication with basis functions maintains spectral structure while allowing flexibility. Moreover, the simple point-wise multiplication design maintains interpretability - we can directly analyze how each spatial location modulates different frequency components, enabling insights into learned spectral processing patterns (see Section 4.6).

While more sophisticated coupling designs are possible, we found this straightforward integration provides effective point-wise adaptivity and stability, outperforming previous methods. We believe developing more advanced instantiations could potentially lead to better performance.

### 3.3. Universal Approximation Property

**Theorem 3.1** (Universal Approximation Property). *Holistic Physics Mixer is a learnable integral neural operator (Kovachki et al., 2023), indicating it possesses universal approximation capability for continuous operator mappings between function spaces. The proof is provided in Section A.1.2 of the Appendix.*

The universal approximation capability provides the theoretical guarantee that Holistic Physics Solver can approximate any continuous operator mapping with arbitrary precision, making it suitable for learning solutions of various PDEs.

### 3.4. Implementation of Holistic Physics Solver

We implement Holistic Physics Mixer based PDE Solver (denoted as HPM) by inserting Holistic Physics Mixer $\mathcal{F}_{\text{Holi}}^{\text{mixer}}$ into existing model architectures (Wu et al., 2024). We used the toolkit of PaddlePaddle to develop this new model and solved the problem.

**Network Architecture.** Following previous works (Wu et al., 2024; Tran et al., 2021), we implement the Holistic Physics Solver as a stack of Holistic Physics Blocks, along with projecting layers $P$ and $V$ at the beginning and end:

$$\mathcal{G}_{\theta}^{\text{Holi}} = V \circ M_l^{Holi} \circ ... \circ M_2^{Holi} \circ M_1^{Holi} \circ P, \quad (16)$$

where each block $M_i^{Holi}$ employs Holistic Physics Mixer

for token mixing:

$$\boldsymbol{v}_{i-1}^{\text{mid}} = \mathcal{F}_{\text{Holi}}^{\text{mixer}}(\text{LayerNorm}(\boldsymbol{v}_{i-1})) + \boldsymbol{v}_{i-1}, \quad (17)$$

$$\boldsymbol{v}_i = \text{FeedForward}(\text{LayerNorm}(\boldsymbol{v}_{i-1}^{\text{mid}})) + \boldsymbol{v}_{i-1}^{\text{mid}}, \quad (18)$$

We instantiate Project in $\mathcal{F}_{\text{Holi}}^{\text{mixer}}$ as LayerNorm$(\cdot)$ followed by an MLP layer. Following Wu et al. (2024), we adopt a multi-head design where features are processed by multiple parallel Holistic Physics Mixers and then concatenated. This enables the model to capture different aspects of the operator mapping simultaneously. The detailed multi-head implementation is described in Section A.2.1.

**Calculation of $\boldsymbol{\Phi}$.** Following Chen et al. (2023), we employ the eigenfunction of Laplace-Beltrami operator as $\boldsymbol{\Phi}$ for its flexibility in handling both regular and irregular domains. Given the discrete representation of physical domain, we use the robust-laplacian library[2] to compute these eigenfunctions (Sharp & Crane, 2020). This allows processing both structured grids and unstructured meshes.

**Discussion.** Holistic Physics Solver presents following advantages: (a) It unifies spectral-based and attention-based approaches in a single representation space, enabling simultaneous modeling of domain-level structure and point-wise adaptivity for complex PDEs. (b) Benefiting from the integration of domain-level spectral priors and point-level flexibility, it not only preserves strong generalization on samples with unseen resolutions and in limited training data scenarios but also presents flexible adaptivity for handling fine-scale local details and utilizing increased training samples. (c) It maintains computational efficiency through a simple coupling mechanism, avoiding attention's quadratic complexity (Wu et al., 2024) while adding minimal overhead to fixed spectral methods (Chen et al., 2023).

## 4. Experiment

### 4.1. Overview

**Baselines.** We compare HPM against a lot of existing neural operators, including both spectral-based and attention-based approaches. For attention methods, we use Transolver (Wu et al., 2024) as the primary baseline due to its leading performance across diverse PDE problems. For spectral methods, we construct SpecSolver as our main spectral baseline by adapting fixed spectral operators into the same modern architecture as Transolver (Wu et al., 2024), ensuring fair comparison by maintaining architecture consistency while preserving the core benefits of spectral approaches.

**Main Results.** Our results lead to following findings:

- HPM consistently outperforms both spectral-based and

---

[2]Robust-laplacian library link: https://github.com/nmwsharp/nonmanifold-laplacian

*Table 1.* Performance comparison on structured mesh problems.

| | Model | Darcy Flow (Regular, Steady) | Airfoil (Irregular, Steady) | Navier-Stokes (Regular, Time) | Plasticity (Irregular, Time) |
|---|---|---|---|---|---|
| *Spectral Methods* | FNO (Li et al., 2020) | 1.08e-2 | - | 1.56e-1 | - |
| | WMT (Gupta et al., 2021) | 8.20e-3 | 7.50e-3 | 1.54e-1 | 7.60e-3 |
| | U-FNO (Wen et al., 2022) | 1.83e-2 | 2.69e-2 | 2.23e-1 | 3.90e-3 |
| | Geo-FNO (Li et al., 2023a) | 1.08e-2 | 1.38e-2 | 1.56e-1 | 7.40e-3 |
| | U-NO (Rahman et al., 2022) | 1.13e-2 | 7.80e-3 | 1.71e-1 | 3.40e-3 |
| | F-FNO (Tran et al., 2021) | 7.70e-3 | 7.80e-3 | 2.32e-1 | 4.70e-3 |
| | LSM (Wu et al., 2023) | 6.50e-3 | 5.90e-3 | 1.54e-1 | 2.50e-3 |
| | NORM (Chen et al., 2023) | 9.71e-3 | 5.44e-3 | 1.15e-1 | 4.39e-3 |
| | SpecSolver | 5.41e-3$_{\pm1.15e\text{-}4}$ | 5.13e-3$_{\pm1.75e\text{-}4}$ | 9.46e-2$_{\pm1.24e\text{-}3}$ | 1.21e-3$_{\pm3.61e\text{-}5}$ |
| *Attention Methods* | Galerkin (Cao, 2021) | 8.40e-3 | 1.18e-2 | 1.40e-1 | 1.20e-2 |
| | HT-Net (Liu et al., 2023b) | 7.90e-3 | 6.50e-3 | 1.85e-1 | 3.33e-2 |
| | OFORMER (Li et al., 2023b) | 1.24e-2 | 1.83e-2 | 1.71e-1 | 1.70e-3 |
| | GNOT (Hao et al., 2023) | 1.05e-2 | 7.60e-3 | 1.38e-1 | 3.36e-2 |
| | FactFormer (Li et al., 2023c) | 1.09e-2 | 7.10e-3 | 1.21e-1 | 3.12e-2 |
| | ONO (Xiao et al., 2023) | 7.60e-3 | 6.10e-3 | 1.20e-1 | 4.80e-3 |
| | Transolver (Wu et al., 2024) | 5.70e-3$_{\pm1.00e\text{-}4}$ | 5.30e-3$_{\pm1.00e\text{-}4}$ | 9.00e-2$_{\pm1.30e\text{-}3}$ | 1.23e-3$_{\pm1.00e\text{-}4}$ |
| | HPM | **4.59e-3**$_{\pm1.94e\text{-}4}$ | **4.72e-3**$_{\pm1.51e\text{-}4}$ | **7.34e-2**$_{\pm9.22e\text{-}4}$ | **8.00e-4**$_{\pm4.58e\text{-}5}$ |

attention-based approaches across various PDE problems (Section 4.2 and 4.3).

- HPM exhibits strong generalization and adaptation capabilities, maintaining robust performance with limited training data while efficiently utilizing additional data when available. It also demonstrates excellent zero-shot generalization to unseen resolutions (Section 4.4).

- Analysis of learned holistic spectral processing patterns reveals meaningful physical insights that could guide fixed spectral operator design (Section 4.6).

- Additional results show that HPM achieves faster convergence during training (Figure 5), requires minimal computational overhead compared to fixed spectral methods (Table 6), and maintains superior performance even with reduced frequency numbers $k$ (Table 9).

### 4.2. Structured Mesh Problems

This section compares HPM with previous neural operators on structured mesh problems, where the physical domains are represented with meshes aligned with standard rectangle grids. For these problems, we implement HPM with LBO eigenfunctions calculated on standard rectangle grids.

**Setup.** (a) Problems. The experimental problems include two regular domain problems Darcy Flow and Navier-Stokes from Li et al. (2020), and two irregular domain problems Airfoil and Plasticity from Li et al. (2023a). Darcy Flow and Airfoil are steady-state solving problems, while Navier-Stokes and Plasticity are time-series solving problems. (b) Metric. Same as previous works (Li et al., 2020; Wu et al., 2024; Gao & Wang, 2023), we use Relative L2 between the predicted results and ground truth (the simulated results) as

the evaluation metric, lower value indicating higher PDE solving accuracy. (c) Baselines. We compare HPM with a lot of neural operators, covering both spectral-based methods and attention-based methods. Section A.3 presents more experimental setup detail.

**Quantitative Comparison.** Table 1 presents the quantitative results. HPM significantly improves the performance over past spectral-based methods LSM (Wu et al., 2023) and NORM (Chen et al., 2023), and outperforms the most performed attention-based method Transolver (Wu et al., 2024). This concludes that holistic spectral processing effectively integrates domain-level spectral prior with point-level adaptivity, leading to better feature learning for operator learning on various problems.

**Qualitative Comparison.** In Figure 3, we visualize the prediction error of HPM and Transolver on different problems. The prediction error of HPM is evidently reduced, especially on physical boundaries and some regions with sharp status changes. This further demonstrates the superior operator learning capability of HPM.

### 4.3. Unstructured Mesh Problems

This section compares HPM with previous works on unstructured mesh problems, where the physical domains are represented with irregular triangle meshes. For handling these problems, we independently calculate LBO eigenfunctions for each problem based on their triangle meshes.

**Setup.** (a) Problems. The evaluated problems include Irregular Darcy, Pipe Turbulence, Heat Transfer, Composite, and Blood Flow from Chen et al. (2023). All problems

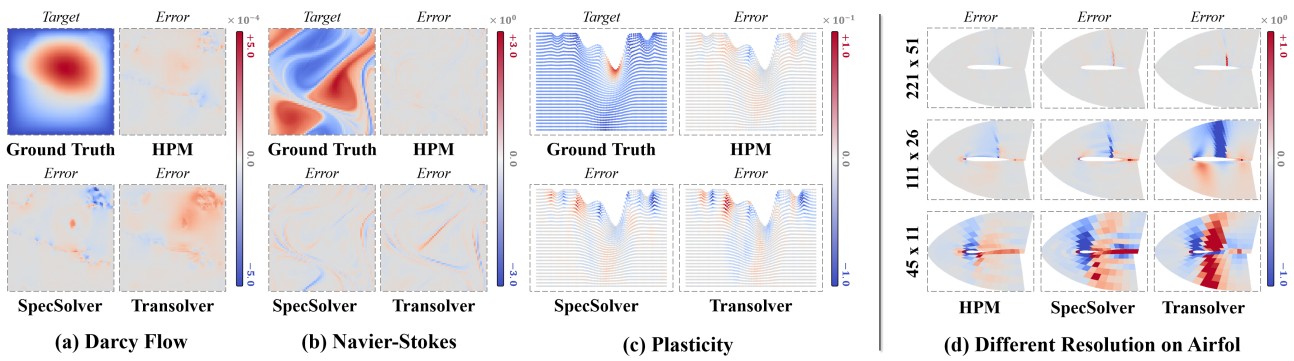

*Figure 3.* Prediction error visualization on different problems.

*Table 2.* Performance comparison on unstructured mesh problems.

| Model | Irregular Darcy (2290 Nodes) | Pipe Turbulence (2673 Nodes) | Heat Transfer (7199 Nodes) | Composite (8232 Nodes) | Blood Flow (1656 Nodes) |
|---|---|---|---|---|---|
| GraphSAGE (Hamilton et al., 2017) | 6.73e-2±5.30e-4 | 2.36e-1±1.41e-2 | - | 2.09e-1±5.00e-4 | - |
| DeepOnet (Lu et al., 2019) | 1.36e-2±1.30e-4 | 9.36e-2±1.07e-3 | 7.20e-4±2.00e-5 | 1.88e-2±3.40e-4 | 8.93e-1±2.37e-2 |
| POD-DeepOnet (Lu et al., 2022) | 1.30e-2±2.30e-4 | 2.59e-2±2.75e-3 | 5.70e-4±1.00e-5 | 1.44e-2±6.00e-4 | 3.74e-1±1.19e-3 |
| FNO (Li et al., 2020) | 3.83e-2±7.70e-4 | 3.80e-2±2.00e-5 | - | - | - |
| NORM (Chen et al., 2023) | 1.05e-2±2.00e-4 | 1.01e-2±2.00e-4 | 2.70e-4±2.00e-5 | 9.99e-3±2.70e-4 | 4.82e-2±6.10e-4 |
| SpecSolver | 7.96e-3±7.19e-5 | 1.11e-2±1.00e-3 | 1.11e-3±3.25e-4 | 1.00e-2±5.24e-4 | 3.73e-2±5.83e-4 |
| HPM | **7.38e-3**±6.20e-5 | **8.26e-3**±7.60e-4 | **1.84e-4**±2.27e-5 | **9.34e-3**±2.71e-4 | **2.89e-2**±3.25e-3 |

come from realistic industry scenarios and include both steady-state problems and time-series problems. (b) Metric. Same as Section 4.2, Relative L2 between the predicted results and ground truth (the simulated results) is used as the evaluation metric, lower value indicating better performance. (c) Baselines. The compared methods include GraphSAGE (Hamilton et al., 2017), DeepOnet (Lu et al., 2019), POD-DeepOnet (Lu et al., 2022), FNO (Li et al., 2020) and NORM (Chen et al., 2023). Section A.3 presents more experimental setup detail.

**Results.** The results are shown in Table 2. Compared to previous methods, HPM obtains consistent enhanced performance across all problems. This validates the benefits of our holistic spectral processing approach that effectively integrates domain-level structure with point-wise adaptivity on complex physical domains and operator mappings.

### 4.4. Generalization Capability Comparison

This section compares the generalization performance of HPM with the attention-based neural operator Transolver (Wu et al., 2024) and the spectral-based SpecSolver.

**Zero-shot Resolution Generalization.** We evaluate the zero-shot capabilities of HPM, Transolver, and SpecSolver on samples with unseen resolutions on Airfoil. The model is trained on the $211 \times 51$ resolution and then tested on lower resolutions including $111 \times 26$ and $45 \times 11$, as well as varied ratio resolutions including $221 \times 26$ and $111 \times 51$. We utilize Relative L2 as the performance metric, with a

*Table 3.* Zero-shot resolution generalization on Airfoil.

| | Resolution | Transolver | SpecSolver | HPM |
|---|---|---|---|---|
| *Training Resolution* | $221 \times 51$ | 5.24e-3 | 5.33e-3 | **4.38e-3** |
| *Consistent Ratio* | $111 \times 26$ | 7.68e-2 | 1.90e-2 | **1.74e-2** |
| | $45 \times 11$ | 9.73e-2 | 7.30e-2 | **5.34e-2** |
| *Inconsistent Ratio* | $221 \times 26$ | 7.85e-2 | 1.91e-2 | **1.69e-2** |
| | $111 \times 51$ | 1.26e-2 | 5.80e-3 | **5.37e-3** |

lower value indicating preferred performance.

Table 3 presents the quantitative comparison results, where significant performance gaps between Transolver and HPM are observed. Additionally, we visualize the prediction error of different resolutions in Figure 3 (d) (more in Figure 8). In contrast to Transolver, HPM apparently diminishes prediction error, particularly on lower-resolution samples.

This indicates that HPM retains remarkable resolution generalization ability like spectral-based methods while maintaining the flexibility of attention mechanisms, thus outperforming both Transolver and SpecSolver.

**Limited Training Numbers.** We additionally evaluate the generalization ability of HPM, Transolver (Wu et al., 2024), and SpecSolver with limited training data amount. Specifically, for Darcy Flow and Navier-Stokes, we train neural operators with 200, 400, 600, 800, and 1000 trajectories respectively, and then test on additional 200 trajectories. Same as other experiments, we use Relative L2 as the performance measure, with a lower value meaning better performance.

*Table 4.* Comparison on different training numbers.

| Problem | Training Number | Transolver | SpecSolver | HPM |
|---|---|---|---|---|
| | 200 | 1.75e-2 | 1.10e-2 | **1.06e-2** |
| | 400 | 1.04e-2 | 7.32e-3 | **6.66e-3** |
| Darcy Flow | 600 | 6.87e-3 | 6.20e-3 | **6.03e-3** |
| | 800 | 6.33e-3 | 5.64e-3 | **4.98e-3** |
| | 1000 | 5.24e-3 | 5.33e-3 | **4.38e-3** |
| | 200 | 3.76e-1 | 1.93e-1 | **1.85e-1** |
| | 400 | 3.14e-1 | 1.48e-1 | **1.26e-1** |
| Navier-Stokes | 600 | 2.87e-1 | 1.21e-1 | **1.17e-1** |
| | 800 | 2.49e-1 | 1.04e-1 | **8.25e-2** |
| | 1000 | 9.60e-2 | 9.34e-2 | **7.44e-2** |

*Table 5.* Comparison of different forms of $\mathbf{H}(\boldsymbol{x}, \boldsymbol{\Phi})$.

| Implementation of $\mathbf{H}(\boldsymbol{x}, \boldsymbol{\Phi})$ | Relative Error |
|---|---|
| Form1: $\text{Softmax}(\text{MLP}(\boldsymbol{x})) \odot \boldsymbol{\Phi}$ | **4.38e-3** |
| Form2: $\text{Sigmoid}(\text{MLP}(\boldsymbol{x})) \odot \boldsymbol{\Phi}$ | 4.86e-3 |
| Form3: $\text{Softmax}(\text{AvgPool}(\text{MLP}(\boldsymbol{x}))) \odot \boldsymbol{\Phi}$ | 5.47e-3 |
| Form4: $\text{Softmax}(\text{MLP}(\boldsymbol{x})) + \boldsymbol{\Phi}$ | 4.69e-3 |
| Form5: $\text{Softmax}(\text{MLP}(\text{Concat}(\boldsymbol{x}, \boldsymbol{\Phi})))$ | 5.24e-3 |

*Table 6.* Comparison of inference time.

| Model | Parameter Count | Inference Time |
|---|---|---|
| SpecSolver | 601,537 | 14.8 ms |
| Transolver | 2,819,521 | 30.9 ms |
| HPM | 751,041 | 17.4 ms |

Table 4 shows the results. HPM outperforms both baselines across all training data quantities. With limited data, HPM matches SpecSolver while significantly outperforming Transolver, showing the benefits of spectral continuity prior. As the data amount increases, HPM leverages its point-level flexibility to fully exploit additional training data and surpass both baselines.

This illustrates that HPM effectively combines domain-level spectral structure with point-wise adaptivity for strong performance across different data regimes.

### 4.5. Additional Comparison

**Different Holistic Instantiations.** We compare several instantiations to find the effective way to fuse spectral structure with point-wise adaptivity under minimal overhead.

The results in Table 5 highlight two crucial coupling insights: First, point-wise processing is essential. As *Form 3* shows, local adaptivity cannot be achieved through global operations. Second, enforcing competition between frequency components via Softmax proves more effective, likely due to explicit frequency trade-offs at each point. We believe developing more sophisticated coupling mechanisms that better balance local flexibility with global spectral structure could potentially achieve better performance.

**Inference Time Comparison.** We compare the inference time of different methods on the Airfoil problem, using a single RTX 3090 with batch size 1. As shown in Table 6, HPM adds minimal overhead over SpecSolver while maintaining fewer parameters and faster inference than attention-based method Transolver (Wu et al., 2024).

### 4.6. Analysis of Learned Modulation Patterns

This section studies how HPM learns to modulate spectral patterns. Benefiting from the straightforward point-wise multiplication design of $\mathbf{H}(\cdot, \cdot)$, we can directly visualize how the model adjusts different spectral components across network layers, providing insights into its learned spectral-physical integration strategies.

**Modulation Pattern Definition.** We use $\mathbf{H}(\boldsymbol{x}, \boldsymbol{\Phi}) = \text{Softmax}(\text{MLP}(\boldsymbol{x})) \odot \boldsymbol{\Phi}$ as coupling function and study how each point modulates different spectral basis functions by visualizing $\mathbf{M}(\boldsymbol{x}) = \text{Softmax}(\text{MLP}(\boldsymbol{x}))$. To show this clearly, for each point, we calculate the difference between how much it uses high frequencies (last half of basis functions) versus low frequencies (first half). This gives us a score from -1 to 1, where positive means stronger high frequencies and negative means stronger low frequencies.

**Pattern Analysis Results.** In Figure.4 (more in Figure.6, 7), we show how a 4-layer HPM for Darcy Flow and an 8-layer HPM for Airfoil problem. We found three main patterns: (a) *Spatially Adaptive Modulation*. Different parts of the space use different patterns - some areas focus on high-order functions, others on low-order ones. This shows that HPM adjusts its processing based on local features. (b) *Layer-wise Evolution*. The patterns change in a clear way through the layers. Early layers generally use uniform patterns across space, middle layers vary more and use more high-order functions where needed, and later layers return to uniform patterns across space. (c) *Physical Feature Enhancement*. At boundaries and areas where solutions change rapidly, the model uses more high-order functions, showing these areas need finer detail for prediction.

**Guiding Fixed Basis Design.** HPM's learned patterns can help improve fixed spectral methods. We test if adding more high-frequency components in the middle layers, based on HPM's layer evolution patterns, could help. As Table 7 shows, manually adding high-frequency components in middle layers works better than adding in early or late layers. This shows that HPM's patterns reveal useful information about how to process spectral features.

In summary, HPM learns patterns that change across both space and network layers, balancing local flexibility with domain spectral structure. These patterns help us understand how HPM works and design better fixed spectral methods.

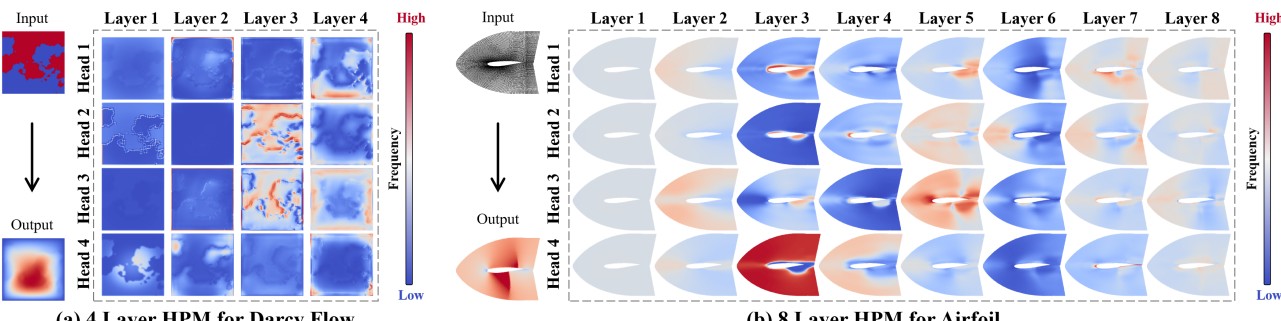

**(a) 4 Layer HPM for Darcy Flow**  **(b) 8 Layer HPM for Airfoil**

*Figure 4.* Visualization of learned holistic spectral modulation patterns on Darcy Flow and Airfoil.

*Table 7.* Fixed basis design guided by HPM's modulation patterns.

| *Basis Enhancement Strategy* | *Relative Error* |
| --- | --- |
| Enhanced High-Freq in Early Layers | 5.12e-3 |
| Enhanced High-Freq in Middle Layers | 4.84e-3 |
| Enhanced High-Freq in Late Layers | 4.97e-3 |
| SpecSolver (No Enhancement) | 5.33e-3 |
| HPM (Adaptive Modulation via $\mathbf{H}(\cdot, \cdot)$) | **4.38e-3** |

## 5. Limitation and Future Work

Despite obtaining superior performance in a lot of scenarios, the introduced Holistic Physics Mixer inevitably suffers certain limitations that do not affect the core conclusion of this work, and they are worth further exploration in future works. (a) Firstly, the current coupling function $\mathbf{H}(\boldsymbol{x}, \boldsymbol{\Phi})$ remains a fundamental design, without considering demands in particular circumstances. Therefore, it is meaningful to develop more sophisticated coupling mechanisms for specific requirements. (b) Additionally, although this work has explored a broad PDE solving problems, numerous real-world physical systems still warrant further investigation. It is significant to investigate the application of Holistic Physics Transform in general deep learning tasks such as time-series signal prediction and computer vision learning.

## 6. Conclusion

This work presents Holistic Physics Mixer based Neural Operators (HPM), which integrates domain-level spectral structure and point-level physical states into a unified representation space for operator learning. The proposed Holistic Physics Transform holds significant potential applications in numerous physics-informed deep learning tasks, as it enables adaptive processing that considers both local physical variations and global spectral properties. Comprehensive experiments validate the superior performance of HPM in various PDE solving scenarios, benefiting from its ability to simultaneously capture domain-level continuity constraints and point-level physical dynamics. We hope HPM's unified treatment of physical systems can inspire future explorations in neural operator learning.

## Acknowledgments

This work is supported by National Science and Technology Major Project (2022ZD0117802). This work was supported in part by General Program of National Natural Science Foundation of China (62372403) and the Earth System Big Data Platform of the School of Earth Sciences, Zhejiang University.

## Impact Statement

This paper presents work whose goal is to advance the field of Machine Learning based PDE solving. There are many potential societal consequences of our work, none of which we feel must be specifically highlighted here.

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

# A. Appendix

## A.1. Theoretical Foundations

We first introduce the preliminary lemmas about neural operator learning in Section A.1.1. Next, we provide the theoretical demonstration (Theory A.4) that Holistic Physics Mixer is the learnable integral neural operator in Section A.1.2.

### A.1.1. PRELIMINARY THEOREM: INTEGRAL NEURAL OPERATOR LEARNING

The following theorems are summarized from previous works (Li et al., 2020; Kovachki et al., 2023; Wu et al., 2024), which provide the theoretical basis of the proposed Holistic Physics Mixer.

**Theorem A.1.** *PDEs could be solved by learning integral neural operators.*

Kovachki et al. (2023) formulate the common architecture of neural operators for PDE solving as a stack of network layers.

$$\mathcal{G}_\theta = Q \circ \sigma(W_l + \mathcal{K}_l) \circ \cdots \circ \sigma(W_i + \mathcal{K}_i) \circ \cdots \circ \sigma(W_1 + \mathcal{K}_1) \circ P, \tag{19}$$

where $P$ and $Q$ are both linear point-wise projectors as shown in Equation 3. $W_i$ is the point-wise fully connected layer and $\mathcal{K}_i$ is the non-local integral operator.

In each network layer, the key is to learn the non-local integral operator $\mathcal{K}_i$ defined as follows:

$$\mathcal{K}_i(\boldsymbol{u})(x) = \int_\Omega \kappa_i(x, \xi, \boldsymbol{u}(x), \boldsymbol{u}(\xi))\boldsymbol{u}(\xi)d\xi, \tag{20}$$

where $\boldsymbol{u}$ is the input function and $\Omega$ is the physical domain. As presented in (Kovachki et al., 2023), the learnable integral kernel operator enables the mapping between continuous functions, similar to the weight matrix operation that enables the mapping between discrete vectors. It could be demonstrated that various neural operators (Li et al., 2020; Cao, 2021; Chen et al., 2023; Wu et al., 2024) are learning different kernel functions of the stacked integral neural operators shown in Equation 20.

**Lemma A.2.** *FNO (Li et al., 2020) learns integral neural operators.*

This is demonstrated in (Li et al., 2020) and (Kovachki et al., 2023). By setting the kernel function as $\kappa(x, \xi, \boldsymbol{u}(x), \boldsymbol{u}(\xi)) = \kappa(x - \xi)$, it could be demonstrated that the kernel integral operator could be implemented with Fourier Transform. For more details you can refer to (Li et al., 2020).

**Lemma A.3.** *The standard Transformer (Vaswani, 2017) learns integral neural operators.*

(Kovachki et al., 2023) demonstrates that the canonical attention mechanism (Vaswani, 2017) is a special case of integral neural operators. This could be demonstrated by setting the kernel function as follows:

$$\kappa(x, \xi, \boldsymbol{u}(x), \boldsymbol{u}(\xi)) = \left(\int_\Omega \exp(\boldsymbol{W}_q\boldsymbol{u}(\xi^{'})(\boldsymbol{W}_k\boldsymbol{u}(x))^T)d\xi^{'}\right)^{-1}\exp(\boldsymbol{W}_q\boldsymbol{u}(x)(\boldsymbol{W}_k\boldsymbol{u}(\xi))^T)\boldsymbol{R}, \tag{21}$$

where $\boldsymbol{W}_q \in \mathbb{R}^{d \times d}$, $\boldsymbol{W}_k \in \mathbb{R}^{d \times d}$ and $\boldsymbol{R} \in \mathbb{R}^{d \times d}$ are all the training parameter of the neural network. For simplification, we eliminate the division operation with $\sqrt{d}$. With this formulation, we can derive the attention mechanism based on the kernel integral operator shown in Equation 20 and Monte-Carlo approximation. The proof can be found in (Kovachki et al., 2023). Therefore, the attention mechanism could be employed for PDE solving.

### A.1.2. HOLISTIC PHYSICS MIXER AS INTEGRAL NEURAL OPERATORS

**Theorem A.4.** *Holistic Physics Mixer is a learnable integral neural operator.*

*Proof.* The Holistic Physics Mixer is represented in the following form:

$$\mathcal{F}_{\text{Holi}}^{\text{mixer}}(\boldsymbol{x}) = \mathcal{T}_{\text{HPT}}^{-1} \circ \text{Project} \circ \mathcal{T}_{\text{HPT}}(\boldsymbol{x}), \tag{22}$$

where $\boldsymbol{x} \in \mathbb{R}^{N \times d}$ is the input feature, $\mathcal{T}_{\text{HPT}}$ and $\mathcal{T}_{\text{HPT}}^{-1}$ represent the Holistic Physics Transform and its inverse transform:

$$\mathcal{T}_{\text{HPT}}(\boldsymbol{x}) = \mathbf{H}(\boldsymbol{x}, \boldsymbol{\Phi})^T\boldsymbol{x}, \tag{23}$$

$$\mathcal{T}_{\text{HPT}}^{-1}(\hat{\boldsymbol{x}}) = \mathbf{H}(\boldsymbol{x}, \boldsymbol{\Phi})\hat{\boldsymbol{x}}. \tag{24}$$

To prove this is an integral neural operator, we construct the kernel function of the integral operator as follows:

$$\kappa(x, \xi, \boldsymbol{u}(x), \boldsymbol{u}(\xi)) = \mathbf{H}(\boldsymbol{x}, \boldsymbol{\Phi})(x)\mathbf{H}(\boldsymbol{x}, \boldsymbol{\Phi})(\xi)^T \boldsymbol{R} \tag{25}$$

where $\mathbf{H}(\boldsymbol{x}, \boldsymbol{\Phi})(x)$ represents the $x$-th row of matrix $\mathbf{H}(\boldsymbol{x}, \boldsymbol{\Phi}) \in \mathbb{R}^{N \times k}$, and $\boldsymbol{R} \in \mathbb{R}^{d \times d}$ is a learnable parameter matrix.

Based on this kernel function, we can derive the Holistic Physics Mixer from the integral neural operator formulation:

$$
\begin{aligned}
\mathcal{K}(\boldsymbol{x})(x) &= \int_{\Omega} \kappa(x, \xi, \boldsymbol{u}(x), \boldsymbol{u}(\xi))\boldsymbol{u}(\xi)d\xi \\
&= \int_{\Omega} \mathbf{H}(\boldsymbol{x}, \boldsymbol{\Phi})(x)\mathbf{H}(\boldsymbol{x}, \boldsymbol{\Phi})(\xi)^T \boldsymbol{R}\boldsymbol{u}(\xi)d\xi \quad \text{(Equation 25)} \\
&= \mathbf{H}(\boldsymbol{x}, \boldsymbol{\Phi})(x)\boldsymbol{R} \int_{\Omega} \mathbf{H}(\boldsymbol{x}, \boldsymbol{\Phi})(\xi)^T \boldsymbol{u}(\xi)d\xi \\
&\approx \mathbf{H}(\boldsymbol{x}, \boldsymbol{\Phi})(x)\boldsymbol{R} \sum_{\xi \in \Omega'} \mathbf{H}(\boldsymbol{x}, \boldsymbol{\Phi})(\xi)^T \boldsymbol{u}(\xi) \quad \text{(Monte-Carlo approximation)} \\
&= \mathbf{H}(\boldsymbol{x}, \boldsymbol{\Phi})(x)\boldsymbol{R}\mathcal{T}_{\text{HPT}}(\boldsymbol{x}) \\
&= \mathcal{T}_{\text{HPT}}^{-1}(\boldsymbol{R}\mathcal{T}_{\text{HPT}}(\boldsymbol{x}))(x) \\
&= (\mathcal{T}_{\text{HPT}}^{-1} \circ \text{Project} \circ \mathcal{T}_{\text{HPT}}(\boldsymbol{x}))(x), \quad \text{(Matrix multiplication as Project)},
\end{aligned}
\tag{26}
$$

where $\Omega'$ is the set of sampled points from domain $\Omega$. The Monte-Carlo approximation requires: (a) The sampling points in $\Omega'$ are sufficiently dense. (b) The coupling function $\mathbf{H}$ is continuous and bounded. Project (including LayerNorm and FC layer) can be represented as matrix multiplication $\boldsymbol{R}$ that is independent of spatial locations.

The final form in Equation 26 is exactly same as Holistic Physics Mixer defined in Equation 22. This concludes that Holistic Physics Mixer is equivalent to an integral neural operator with kernel function defined in Equation 25. $\qquad\square$

## A.2. Methodology Extension

### A.2.1. MULTI-HEAD HOLISTIC PHYSICS MIXER

Following the multi-head attention mechanism (Vaswani, 2017; Wu et al., 2024), we enhance the Holistic Physics Mixer by introducing a multi-head architecture that processes features in parallel holistic spectral spaces. Specifically, we first split the latent features $\boldsymbol{x} \in \mathbb{R}^{N \times d_v}$ into $h$ vectors $\boldsymbol{x}^{\text{head-1}}$, $\boldsymbol{x}^{\text{head-2}}$, ..., $\boldsymbol{x}^{\text{head-h}}$ along the channel dimension, where $\boldsymbol{x}^{\text{head-i}} = \boldsymbol{x}_{[:, d_v^{\text{head}} \times (i-1): d_v^{\text{head}} \times i]}$ and $h$ denotes the number of heads. $d_v^{\text{head}} = d_v/h$ is the dimension of features in single head.

Next, every vector $\boldsymbol{x}^{\text{head-i}} \in \mathbb{R}^{N \times d_v^{\text{head}}}$ is independently processed by $\mathcal{F}_{\text{Holi}}^{\text{mixer}}$. The Holistic Physics Mixer in each head is formulated as:

$$\mathcal{F}_{\text{Holi}}^{\text{mixer}}(\boldsymbol{x}) = \mathcal{T}_{\text{HPT}}^{-1} \circ \text{FC} \circ \text{LayerNorm} \circ \mathcal{T}_{\text{HPT}}(\boldsymbol{x}), \tag{27}$$

where LayerNorm$(\cdot)$ is introduced to normalize the holistic spectral features for more efficient optimization and enhanced generalization. Additionally, we share the learnable weights of FC for all holistic spectral components.

Finally, all output vectors are concatenated as the final output:

$$\mathcal{F}_{\text{Holi}}^{\text{multi-head-mixer}}(\boldsymbol{x}) = \text{Concat}(\mathcal{F}_{\text{Holi}}^{\text{mixer}}(\boldsymbol{x}^{\text{head-i}})). \tag{28}$$

The multi-head design brings several benefits:

- Each head can learn different coupling patterns between spectral and physical information, allowing the model to capture various aspects of the operator mapping simultaneously.
- The parallel computation of multiple heads enables efficient implementation on modern hardware.
- The concatenation of multiple heads provides richer feature representations that combine different aspects of the holistic spectral space.

In practice, we find that using 4-8 heads typically provides good performance across different PDE problems. The specific head numbers used for each problem are reported in Table 8.

A.2.2. SPARSE-FREQUENCY FIXED SPECTRAL TRANSFORM

To enhance the learning capability of fixed spectral methods, we attempt to manually add high-frequency spectral features in several network layers based on insights gained from HPM's learned coupling patterns. Specifically, instead of using the lowest $k$ frequencies, we uniformly take $k$ frequencies from the lowest $k \times r$ frequencies, where $r$ is the sparsity ratio. Higher $r$ indicates using more high frequencies and we set $r = 2$ and $r = 4$ for different layers.

We find that using Sparse-Frequency Spectral Transform in partial network layers effectively improves the performance of fixed spectral methods. However, such manual frequency design relies on prior knowledge and repeated experiments to select appropriate layers and sparsity ratios, and additional computational cost is required for calculating LBO eigenfunctions with $k \times r$ frequencies. To address this issue, we experiment with the fixed spectral design guided by learned coupling patterns of HPM, as shown in Table 7.

**A.3. Experiment Setups**

A.3.1. IMPLEMENTATION DETAIL

We implement HPM with comparable parameter count to the compared baselines (Hao et al., 2023; Wu et al., 2024; Chen et al., 2023). The same optimizer setup as Transolver (Wu et al., 2024) is employed. All experiments (including all baselines, ablations and our method) could be conducted with a single A100 device. The implementation detail for each problem is presented in Table 8.

A.3.2. METRIC

Same as previous works (Li et al., 2020; Wu et al., 2024), the assessed metric in this work is the Relative L2 Error, formulated as follows:

$$L2 = \frac{1}{N_{\text{test}}} \sum_{i=1}^{N_{\text{test}}} \frac{\|\hat{u}_i - u_i\|_2}{\|u_i\|_2}, \tag{29}$$

where $N_{\text{test}}$ is the number of evaluated samples, $\hat{u}_i$ represents the predicted trajectory, and $u_i$ denotes the ground-truth trajectory.

A.3.3. EVALUATED PDE PROBLEMS

**Darcy Flow.** Darcy Flow is a steady-state solving problem from Li et al. (2020). We experiment with the identical setup as previous works (Li et al., 2020; Tran et al., 2021; Wu et al., 2024). The resolution of input and output functions are $85 \times 85$ and there are 1000 trajectories for training and an additional 200 data for testing.

**Navier-Stokes.** Navier-Stokes is the PDE solving problem introduced in FNO (Li et al., 2020). We experiment with the most challenging split where the viscosity coefficient is 1e-5. The input is the vorticity field of the first 10 time steps and the target is to predict the status of the following 10 steps. The training and test amounts are 1000 and 200 respectively.

**Airfoil.** Airfoil is an irregular domain problem from Geo-FNO (Li et al., 2023a). In this experiment, the neural operators take the airfoil shape as input and predict the Mach number on the domain. The irregular domain is represented as structured meshes aligned with standard rectangles. All airfoil shapes come from the NACA-0012 case by the National Advisory Committee for Aeronautics. 1000 samples are used for training and additional 200 samples are used for evaluation.

**Plasticity.** This task requires neural operators to predict the deformation state of plasticity material and the impact from the upper boundary by an irregular-shaped rigid die. The input is the shape of the die and the output is the deformation of each physical point in four directions in future 20 time steps. There are 900 data for training and an additional 80 data for testing.

**Irregular Darcy.** This problem involves solving the Darcy Flow equation within an irregular domain. The function input is $a(x)$, representing the diffusion coefficient field, and the output $u(x)$ represents the pressure field. The domain is represented by a triangular mesh with 2290 nodes. The neural operators are trained on 1000 trajectories and tested on an extra 200 trajectories.

**Pipe Turbulence.** Pipe Turbulence system is modeled by the Navier-Stokes equation, with an irregular pipe-shaped computational domain represented as 2673 triangular mesh nodes. This task requires the neural operator to predict the next frame's velocity field based on the previous one. Same as Chen et al. (2023), we utilize 300 trajectories for training and

*Table 8.* Implementation detail for each PDE problem.

| Problems | Model Configurations | | | | Training Configurations | | | | | |
| --- | --- | --- | --- | --- | --- | --- | --- | --- | --- | --- |
| | Depth | Width | Head Number | $k$ | Optimizer | Scheduler | Initial Lr | Weight Decay | Epochs | Batch Size |
| Darcy Flow | 8 | 128 | 8 | 128 | AdamW | OneCycleLR | 1e-3 | 1e-5 | 500 | 4 |
| Airfoil | 8 | 128 | 8 | 128 | AdamW | OneCycleLR | 1e-3 | 1e-5 | 500 | 4 |
| Navier-Stokes | 8 | 256 | 8 | 128 | AdamW | OneCycleLR | 1e-3 | 1e-5 | 500 | 4 |
| Plasticity | 8 | 128 | 8 | 128 | AdamW | OneCycleLR | 1e-3 | 1e-5 | 500 | 8 |
| Irregular Darcy | 4 | 64 | 4 | 64 | AdamW | OneCycleLR | 1e-3 | 1e-5 | 2000 | 16 |
| Pipe Turbulence | 4 | 64 | 4 | 64 | AdamW | OneCycleLR | 1e-3 | 1e-5 | 2000 | 16 |
| Heat Transfer | 4 | 64 | 4 | 64 | AdamW | OneCycleLR | 1e-3 | 1e-5 | 2000 | 16 |
| Composite | 4 | 64 | 4 | 64 | AdamW | OneCycleLR | 1e-3 | 1e-5 | 2000 | 16 |
| Blood Flow | 4 | 64 | 4 | 32 | AdamW | OneCycleLR | 1e-3 | 1e-5 | 2000 | 4 |

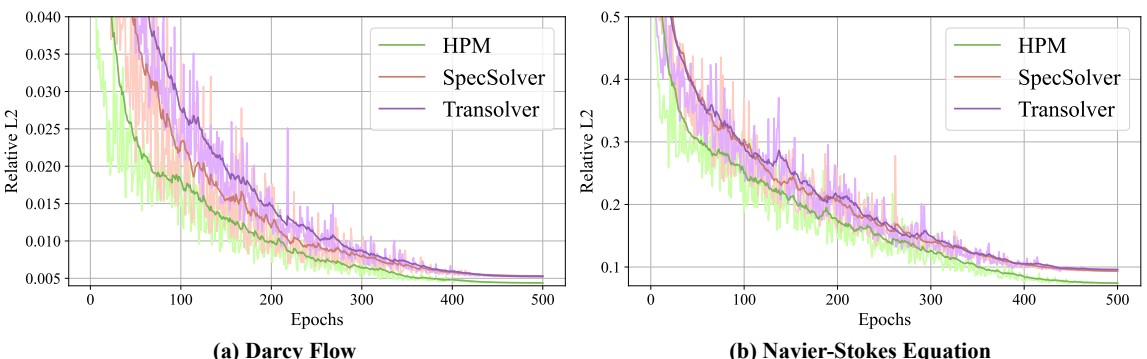

**(a) Darcy Flow**    **(b) Navier-Stokes Equation**

*Figure 5.* Comparison of validation loss curve during training.

then test the models on 100 samples.

**Heat Transfer.** This problem is about heat transfer events triggered by temperature variances at the boundary. Guided by the Heat equation, the system evolves over time. The neural operator strives to predict 3-dimensional temperature fields after 3 seconds given the initial boundary temperature status. The output domain is represented by triangulated meshes of 7199 nodes. The neural operators are trained on 100 data sets and evaluated on another 100 data.

**Composite.** This problem involves predicting deformation fields under high-temperature stimulation, a crucial factor in composite manufacturing. The trained operator is anticipated to forecast the deformation field based on the input temperature field. The structure studied in this paper is an air-intake component of a jet composed of 8232 nodes, as referenced in (Chen et al., 2023). The training involved 400 data, and the test examined 100 data.

**Blood Flow.** The objective is to foresee blood flow within the aorta, including 1 inlet and 5 outlets. The flow of blood is deemed a homogeneous Newtonian fluid. The computational domain, entirely irregular, is visualized by 1656 triangle mesh nodes. Over a simulated 1.21-second duration, with 0.01-second temporal steps, the neural operator predicts different times' velocity fields given velocity boundaries at the inlet and pressure boundaries at the outlet. Same as (Chen et al., 2023), our experiment involves training on 400 data sets and testing on 100 data.

### A.4. Additional Experimental Results

This section ablates the core modules of HPM to reveal the main factors affecting its performance.

**Optimization Efficiency Comparison.** In addition, we compare the validation loss curves of HPM and Transolver during training, as portrayed in Figure 5. We notice that HPM reaches the same prediction accuracy as Transolver earlier, often dozens or even hundreds of epochs ahead, especially in the initial and middle stages of training (the first 300 epochs). This confirms the excellent operator fitting ability of HPM benefiting from the suitable combination of spectral priors (offering fundamental function approximation basis) and point-wise adaptivity (providing efficient local feature processing).

**Frequency Number.** We compare the performance of HPM and SpecSolver with different frequency numbers $k$, as shown in Table 9. HPM consistently performs better than SpecSolver across all frequency settings. The performance gap is particularly significant under lower frequency numbers (16, 32, and 64). This leads to the conclusion that holistic spectral

*Table 9.* Performance of different frequency numbers on Darcy Flow.

| Problem | Frequency Number | SpecSolver | HPM |
|---|---|---|---|
| Darcy Flow | 16 | 1.04e-2 | **5.10e-3** |
| | 32 | 8.08e-3 | **5.02e-3** |
| | 64 | 6.15e-3 | **4.85e-3** |
| | 128 | 5.31e-3 | **4.38e-3** |
| Navier-Stokes | 16 | 1.18e-1 | **9.51e-2** |
| | 32 | 1.05e-1 | **8.77e-2** |
| | 64 | 9.47e-2 | **7.44e-2** |
| | 128 | 8.37e-2 | **7.28e-2** |

processing eliminates the dependency on a large number of frequencies, benefiting from the point-wise adaptive mechanism. Therefore, HPM potentially performs better in some practical industry scenarios where computing high-frequency basis functions is computationally expensive.

**Head Number Analysis.** We evaluate the impact of varying head numbers in HPM on the Darcy Flow problem. As shown in Table 10, using multiple heads (4, 8, or 16) improves performance compared to a single head. However, HPM remains relatively insensitive to the specific number of heads chosen, with all multi-head variants achieving similar performance levels.

*Table 10.* Impact of head numbers.

| Head Number | Relative Error |
|---|---|
| 1 | 4.77e-3 |
| 2 | 4.79e-3 |
| 4 | 4.56e-3 |
| 8 | 4.38e-3 |
| 16 | 4.54e-3 |

**Impact of Parameters.** To validate that HPM's performance gains come from its architecture rather than increased parameters, we evaluate HPM with reduced depth on the Darcy Flow problem.

*Table 11.* Performance comparison with different parameter amounts.

| Model | Layers | Parameter Amount | L2 Error |
|---|---|---|---|
| Transolver | 8 | 2,835,649 | 5.24e-3 |
| SpecSolver | 8 | 617,665 | 5.31e-3 |
| HPM | 8 | 767,169 | **4.38e-3** |
| HPM | 4 | 408,737 | 4.71e-3 |

As shown in Table 11, HPM with only 4 layers still outperforms both Transolver (Wu et al., 2024) and SpecSolver, despite having significantly fewer parameters. This demonstrates that the performance gains come from the holistic spectral processing architecture rather than increased model capacity.

**Additional Resolution Generalization Experiment.** To further evaluate resolution generalization capabilities, we train models on a middle resolution ($111 \times 26$) and test on both higher ($221 \times 51$) and lower ($45 \times 11$) resolutions on the Airfoil problem.

As shown in Table 12, HPM maintains strong performance across both upsampling and downsampling scenarios. While attention-based methods like Transolver excel at training resolution, they struggle to generalize to unseen resolutions. In contrast, HPM combines high accuracy at the training resolution with robust generalization capabilities inherited from spectral methods.

_Table 12._ Resolution generalization performance on Airfoil.

| Model | Train Resolution | Higher Resolution | Lower Resolution |
|---|---|---|---|
| Transolver | 4.50e-3 | 6.95e-2 | 1.22e-1 |
| SpecSolver | 5.37e-3 | 1.95e-2 | 5.84e-2 |
| HPM | **4.17e-3** | **1.57e-2** | **3.85e-2** |

## A.5. Additional Visualization of Point-wise Frequency Preference

**Visualization of Modulation Patterns for Samples with Different Resolutions.** Figure 6 presents additional visualizations of holistic spectral modulation patterns on the Airfoil problem for samples with varied resolutions. We observe that the learned modulation patterns remain consistent as the domain resolution changes. This demonstrates that HPM's coupling mechanism maintains stable frequency preferences across different resolutions, contributing to its strong resolution generalization capability.

**Visualization of Modulation Patterns for Different Samples.** Figure 7 shows the visualization of learned modulation patterns for different samples on Darcy Flow. Despite handling different input functions, each head maintains consistent modulation strategies across samples at specific layers. For example, Head-1 in Layer-1 consistently enhances high-frequency components near boundaries, while Head-1 in Layer-2 emphasizes high frequencies in regions with sharp state changes. This indicates that the multi-head architecture in HPM learns specialized modulation strategies, with different heads focusing on distinct aspects of spectral-physical coupling. These systematic patterns reveal how HPM combines domain-level spectral structure with point-wise adaptivity to achieve effective operator learning.

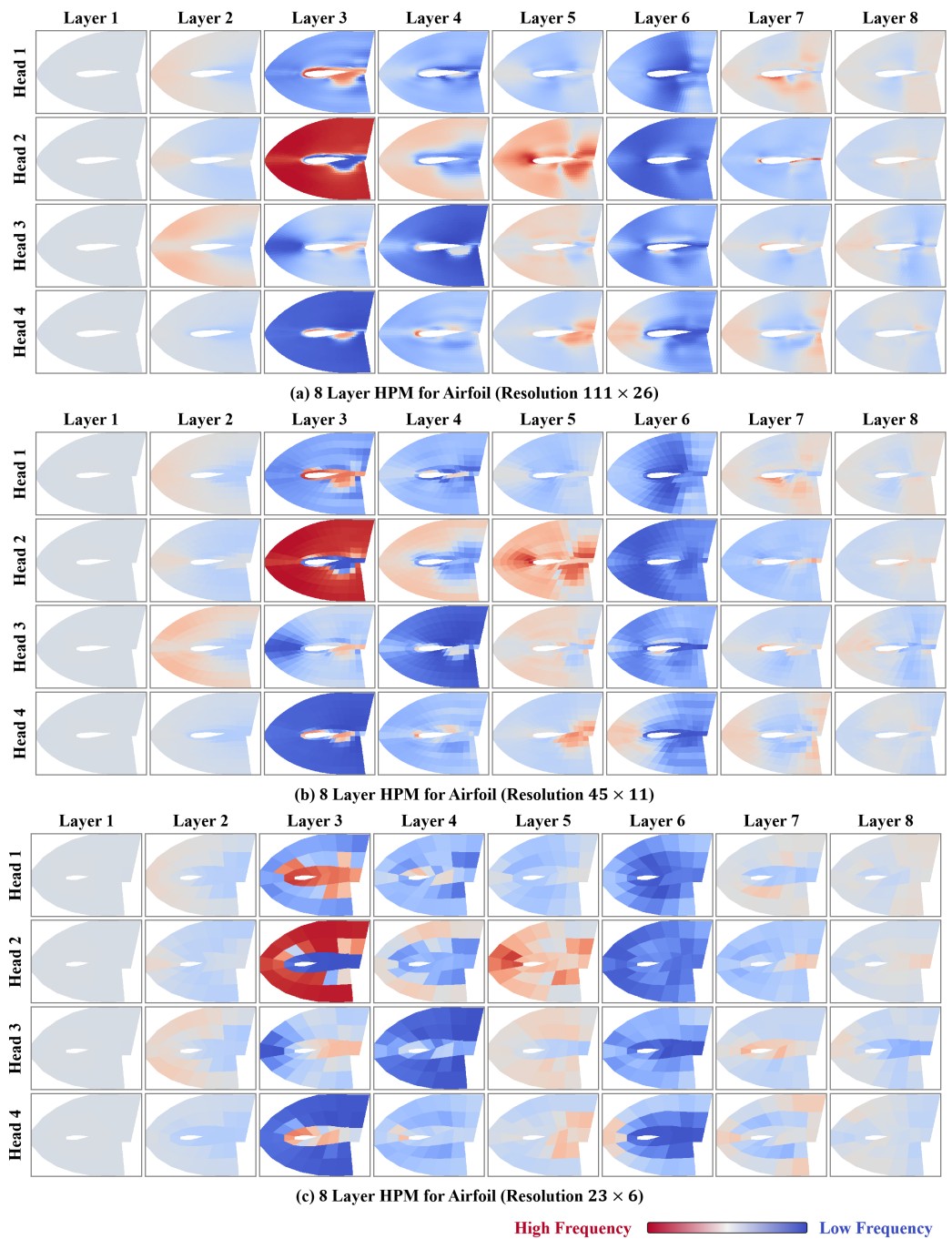

*Figure 6.* Visualization of learned holistic spectral modulation patterns on Airfoil for samples with different resolutions.

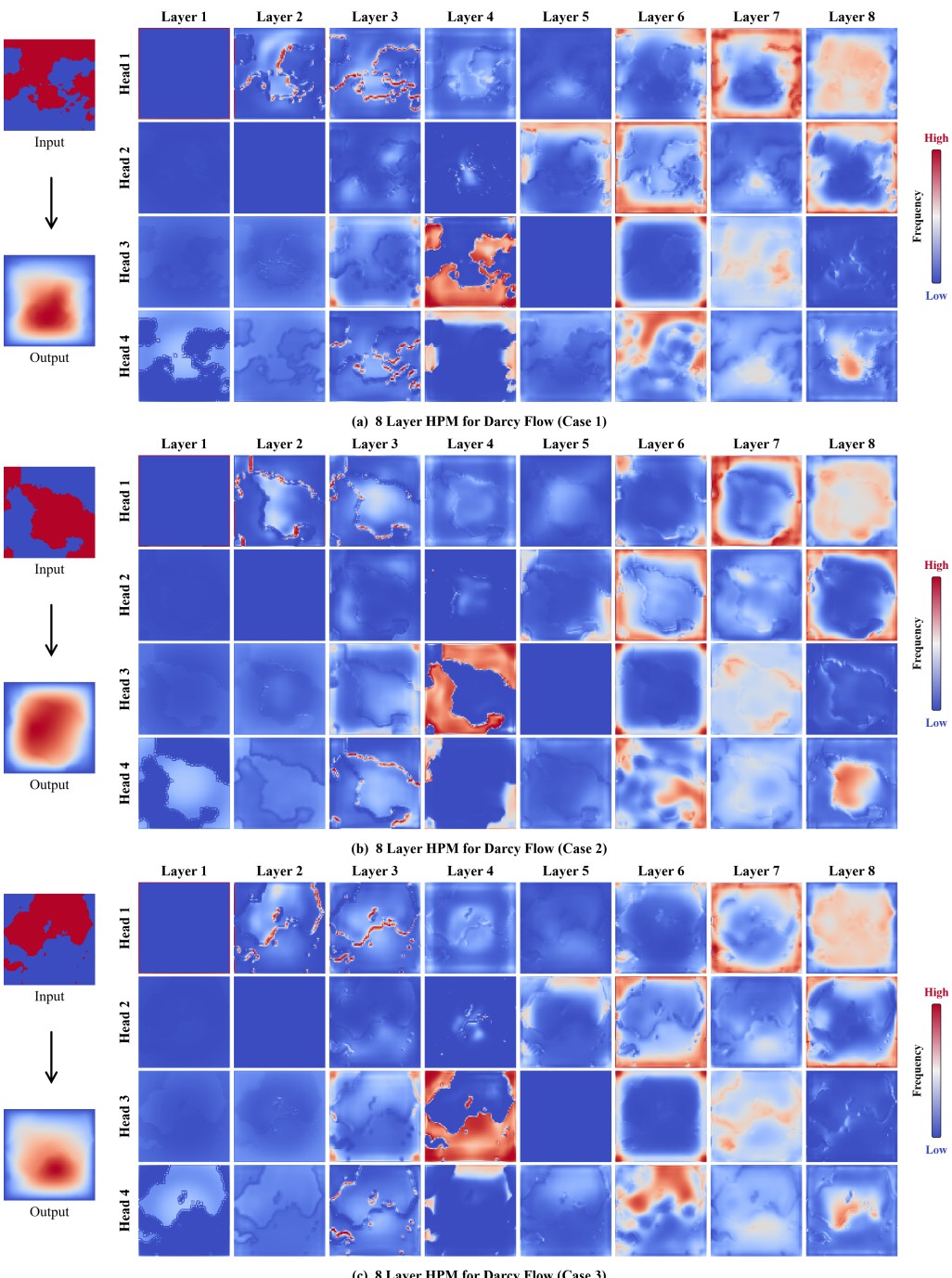

*Figure 7.* Visualization of learned holistic spectral modulation patterns on Darcy Flow for different samples.

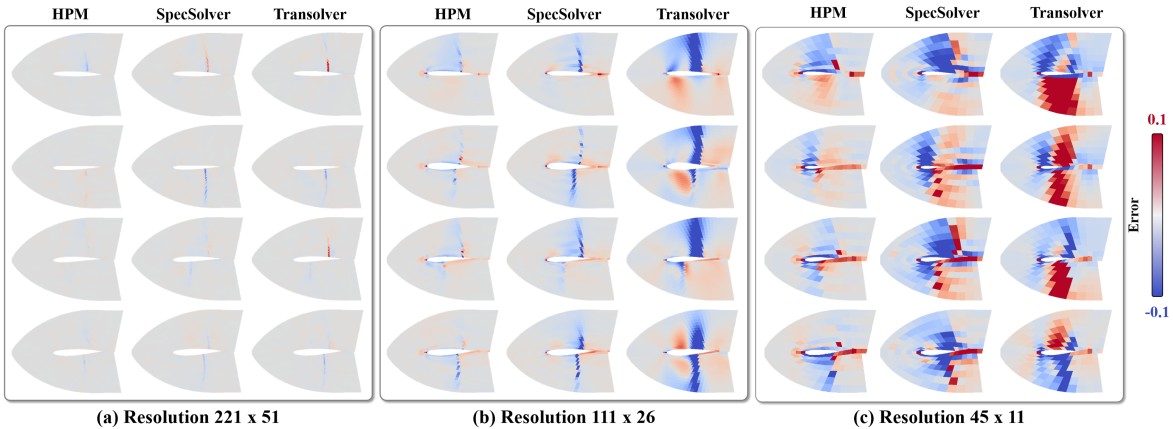

*Figure 8.* More visualization of prediction error on different test resolutions.

