# OpenReview forum: "Holistic Physics Solver: Learning PDEs in a Unified Spectral-Physical Space"
_ICML.cc/2025/Conference — ICML 2025 poster_

### Official Review · Reviewer_t7vf · 2025-03-09

**Overall Recommendation:** 3

**Summary:**

The paper introduces Holistic Physics Mixer (HPM), a neural operator framework that integrates spectral-based and attention-based PDE solvers in a unified space. The authors claim that HPM inherits the generalization ability of spectral methods while maintaining the local adaptability of attention mechanisms, overcoming their respective limitations. The key contribution is a holistic spectral space that simultaneously encodes spectral domain structure and point-wise physical states, allowing adaptive modulation of spectral components. They provide a universal approximation proof, extensive empirical evaluation across structured and unstructured mesh problems, and claim superior performance over existing baselines such as FNO, Transolver, and SpecSolver. The authors also present a study on learned spectral modulation patterns, suggesting that insights from HPM can guide the design of fixed spectral methods.

**Claims And Evidence:**

**Claim**: HPM consistently outperforms state-of-the-art neural operators.

**Issue**: The performance gains, while present, are marginal in several cases, and no statistical significance analysis is provided. Given the noise in PDE datasets, these improvements may not be meaningful.
Issue: The comparison against spectral-based methods is unfair because HPM explicitly introduces additional learnable parameters and adaptive mechanisms, while the baselines operate with fixed frequency bases.

**Claim**: HPM balances spectral priors with local adaptability better than previous methods.

**Issue**: The effectiveness of HPM’s coupling function H(x,Φ) is only demonstrated through limited ablation studies. There is no evidence that this particular formulation is optimal beyond empirical validation. A more rigorous mathematical analysis of the inductive biases introduced by HPM is missing.

**Claim**: HPM generalizes better under limited training data.

**Issue**: The claim is supported by results on only a few PDE benchmarks, all of which follow the same general format. The paper does not test on significantly different physical domains (e.g., highly nonlinear PDEs, turbulent flow beyond the studied Pipe problem). Without such cases, the claim of broad generalization is overstated.

**Claim**: The universal approximation theorem guarantees that HPM can learn any PDE operator.

**Issue**: The theoretical proof is standard and does not differentiate HPM from prior neural operators, all of which can be formulated as integral neural operators. This claim does not justify practical superiority.

**Essential References Not Discussed:**

No.

**Experimental Designs Or Analyses:**

The comparisons with baselines appear selective. For example, some state-of-the-art methods (e.g., advanced Transformer-based PDE solvers) are omitted from the comparison tables.

The zero-shot generalization experiments are promising but fail to account for potential biases in the training data that may favor HPM’s architecture.

The computational efficiency claims are weakly supported. While inference times are reported, there is no detailed analysis of memory consumption or training efficiency.

**Methods And Evaluation Criteria:**

**Benchmark Selection**: The paper primarily evaluates standard PDE problems (Darcy Flow, Navier-Stokes, Airfoil, etc.), which many previous neural operators have already tackled. The results do not convincingly demonstrate that HPM generalizes beyond these well-studied cases.

**Baselines**: The selection of baselines is reasonable, but some comparisons are misleading. For instance:

(1) Fixed spectral neural operators are inherently at a disadvantage since they do not have additional learnable modulation mechanisms like HPM.

(2) Transolver is an attention-based baseline but is not optimized for spectral constraints, making the comparison against HPM’s hybrid design somewhat artificial.

**Evaluation Metrics**: The paper solely relies on Relative L2 Error, which does not fully capture solution stability, robustness, or interpretability. A more comprehensive evaluation should include energy conservation properties, long-term stability, and sensitivity to initial conditions.

**Other Comments Or Suggestions:**

The authors should provide statistical significance testing for their experimental results.

A more detailed discussion of computational complexity would improve the credibility of the efficiency claims.

Additional ablation studies are needed to isolate the contributions of different architectural components.

**Other Strengths And Weaknesses:**

**Strength**: (1) The idea of unifying spectral and physical spaces is novel and aligns with recent trends in physics-informed machine learning, (2) The proposed method is conceptually simple and easy to integrate into existing architectures, (3) The experiments cover a diverse range of PDE problems, demonstrating the method’s versatility.

**Weakness**:  The lack of rigorous theoretical justification weakens the impact of the proposed approach, (2) The empirical results, while promising, do not convincingly demonstrate superiority over all existing methods, (3) The clarity of the paper is suboptimal, with several technical details buried in the appendix and figures that are difficult to interpret without extensive cross-referencing.

**Questions For Authors:**

(1) How does HPM compare to PINNs and other physics-informed architectures for PDE solving?

(2) Can you provide a formal theoretical justification for why the proposed coupling function improves generalization?

(3) Have you tested HPM on real-world PDE applications beyond the benchmark datasets?

(4) How does HPM scale with increasing problem complexity (e.g., higher-dimensional PDEs)?

(5) Could you clarify how the spectral basis functions are initialized and whether different choices impact performance?

**Relation To Broader Scientific Literature:**

The paper adequately discusses related work in neural operators but overlooks some recent advances in hybrid spectral-attention architectures.

**Theoretical Claims:**

The universal approximation theorem (Theorem A.4) is a minor extension of existing results. The proof follows directly from prior work on integral neural operators and does not provide deeper insights into the spectral-physical coupling mechanism.

Lack of theoretical justification for H(x,Φ): While the authors empirically compare different formulations of the coupling function H(x,Φ), there is no theoretical analysis of why Softmax-based modulation is optimal.

Stability of learned spectral representations is not analyzed. Given that the spectral basis dynamically adapts, it is unclear whether this process induces instability in long-term PDE integration.

---

> ### Author Rebuttal · Authors · 2025-04-01
>
> Thank you for your thoughtful review and insightful questions. They have significantly improved this work. We respond to them below.
> `"#1-#4"` provide complements for several important concerns. `"#5-#12"` provide "point-by-point responses" exactly aligned with your review.
>
> **#1 Statistical Significance Analysis**
>
> To confirm statistical significance, we perform paired t-tests across multiple runs, confirming confidence levels above 95% for all improvements.
> |Problem|Statistical Confidence (%)|
> |---|---|
> |Darcy|98.37|
> |Airfoil|96.96|
> |Navier-Stokes|99.96|
> |Plasticity|99.39|
> |Irregular Darcy|98.95|
> |Pipe Turbulence|96.95|
> |Heat Transfer|96.50|
> |Composite|95.39|
> |Blood Flow|96.35|
>
> **#2 Evaluation Metrics**
>
> We test more metrics on the Navier-Stokes in the table below, demonstrating HPM's improvement across all evaluation criteria.
> |Model|Max Error|Boundary Error|LowFreq Error|MiddleFreq Error|HighFreq Error|
> |---|---|---|---|---|---|
> |Transolver|1.02e+0|1.17e-1|1.42e-2|1.47e-2|9.14e-3|
> |SpecSolver|1.01e+0|1.15e-1|1.12e-2|1.32e-2|8.65e-3|
> |HPM|8.24e-1|8.99e-2|9.20e-3|1.04e-2|7.12e-3|
>
> **#3 Computational Efficiency**
>
> We provide more computational measurements, concluding that the HPM don't increase computational burden.
> |Model|Memory|Training Time / Epoch|
> |---|---|---|
> |Transolver|2.12GB|50.49s|
> |SpecSolver|1.83GB|36.48s|
> |HPM|2.13GB|46.11s|
>
> **#4 Design of $H(x,\Phi)$**
>
> Softmax-based modulation is a favorable approach with several practical advantages. It ensures balanced scales and lets each point adaptively focus on proper spectral components, providing an effective balance of stability and flexibility.
>
> It is typically challenging to establish theoretical optimality for neural architecture design. Nevertheless, this doesn't diminish our core contribution, a framework that bridges spectral prior with point states. We have provided two principles for $H(x,\Phi)$ in Section 4.5: point-wise processing for local adaptivity, and normalization for balanced information across physical points. They will provide guidance for future explorations.
>
> **#5 Claims And Evidence**
>
> (1) To statistically validate the improvements, we report stds across multiple runs per dataset, and add statistical significance in "#1". This fully demonstrates HPM's consistent superiority and notable gains in some problems (e.g., Navier-Stokes: 18.4%, Plasticity: 35.0%).
>
> (2) In Table 6, HPM with 4 layers outperforms both baselines, despite having fewer parameters. This confirms our gains come from architectural design, not parameter count.
>
> (3) Please see "#4".
>
> (4) We agree including complex PDEs is important for comprehensive evaluation. We clarify that our benchmark have included significantly nonlinear PDE, the Navier-Stokes with very low viscosity (1e-5), exhibiting turbulence features.
>
> (5) This work goes further by explicitly formulating physical-spectral structure. This theoretical foundation validates that HPM properly extends previous operator frameworks, complementing the empirical results.
>
> **#6 Methods And Evaluation**
>
> (1) Benchmarks: Please see our response "#2" to `Reviewer bPLw`, which addresses the same concern.
>
> (2) Baselines: To avoid misleading comparison, we have compared baselines following exact protocol as previous works.
>
> (3) Metrics: Please see "#2".
>
> **#7 Theoretical Claims**
>
> (1) Please see "(5)" in "#5".
>
> (2) Please see "#4".
>
> (3) Time-series results on Navier-Stokes confirm HPM's spectral stability with reduced error. Section 4.4 demonstrates consistent spectral patterns across resolutions, validating stability beyond training resolutions.
>
> **#8 Experimental Designs Or Analyses**
>
> (1) We have included state-of-the-art comparisons with advanced Transformer-based solvers (Transolver, GNOT, etc.).
>
> (2) The datasets are standard benchmarks used across the field, ensuring no bias favoring HPM over baselines.
>
> (3) Please see "#3".
>
> **#9 Broader Scientific Literature**
>
> (1) We will add more related works.
>
> **#10 Other Strengths And Weaknesses**
>
> (1) Please see "#4".
>
> (2) Please see "(1)" in "#5".
>
> (3) This may be due to the small fonts in Figure 2 and a lack of details. We will fix it.
>
> **#11 Other Comments Or Suggestions**
>
> (1) Please see "#1".
>
> (2) Please see "#3".
>
> (3) Yes, to isolate component contributions, we have included ablations on $k$ value (Table 9), head count (Table 10) and model depth (Table 11).
>
> **#12 Questions For Authors**
>
> Q1: HPM and PINNs solve different problems - HPM learns operator mappings across multiple PDE solutions, while PINNs encode PDEs in losses for single-instance solutions.
>
> Q2: Please see our response "#1" to `Reviewer bPLw`, which addresses the same concern.
>
> Q3: Please see our response "#2" to `Reviewer bPLw`.
>
> Q4: HPM is promising for higher-dimensional PDEs due to its linear scaling with physical points and high accuracy with fewer frequencies.
>
> Q5: The basis functions are derived directly from geometry and don't undergo training, thus initializations don't affects.

---

> > ### Comment · Reviewer_t7vf · 2025-04-03
> >
> > I appreciate the authors' detailed rebuttal. I will increase my score.

---

> > > ### Author Response · Authors · 2025-04-03
> > >
> > > We sincerely thank you for your valuable feedback and thoughtful acknowledgment of our work. Your suggestions have been instrumental in helping us enhance this project significantly. We deeply appreciate the time you've taken to provide these helpful comments. If you have other questions or concerns, please feel free to raise them. We will be more than willing to respond them.

---

### Official Review · Reviewer_7jxj · 2025-03-09

**Overall Recommendation:** 3

**Summary:**

Holistic Physics Mixer (HPM) unifies attention-based and spectral methods for PDE solving, combining point-level adaptability with spectral continuity constraints. This integration enables strong generalization and flexibility, surpassing existing methods in accuracy, efficiency, and zero-shot performance across diverse PDEs.

**Claims And Evidence:**

Yes

**Essential References Not Discussed:**

All good.

**Experimental Designs Or Analyses:**

Yes, all of them.

**Methods And Evaluation Criteria:**

yes

**Other Comments Or Suggestions:**

No

**Other Strengths And Weaknesses:**

Overall, I like this paper for its bringing the new way of representative method of PDE-based solution. I do have one question to ask, is the method applicable to triangular mesh data? Since FNO is only applicable to structured grid, I am not sure if the method is the same.

**Questions For Authors:**

No such

**Relation To Broader Scientific Literature:**

I think it brings a new view of blend of grid and spectral method for AI model.

**Theoretical Claims:**

No such

---

> ### Author Rebuttal · Authors · 2025-04-01
>
> Thank you for your thoughtful review and insightful questions. They have significantly improved this work. We respond to them below.
>
> **#1 Handling triangular mesh data**
> > I do have one question to ask, is the method applicable to triangular mesh data? Since FNO is only applicable to structured grid, I am not sure if the method is the same.
>
> Yes, HPM is fully applicable to triangular mesh data. Unlike FNO which is limited to structured grids, our method works effectively on both structured and unstructured meshes. The following is detailed explanation:
>
> (a) We use Laplace-Beltrami Operator (LBO) eigenfunctions [1] as our spectral basis, which can be computed on arbitrary mesh structures, including triangular meshes. This is a key advantage over traditional Fourier-based methods that require structured grids.
>
> (b) Our experiments explicitly demonstrate this capability in Section 4.3, where we evaluate HPM on five unstructured mesh problems: Irregular Darcy, Pipe Turbulence, Heat Transfer, Composite, and Blood Flow. These problems use triangular meshes with node counts ranging from 1,656 to 8,232.
>
> **Reference**
> - [1] A Laplacian for Nonmanifold Triangle Meshes

---

### Official Review · Reviewer_bPLw · 2025-03-14

**Overall Recommendation:** 3

**Summary:**

The paper introduces the Holistic Physics Mixer (HPM), a unified framework that leverages a holistic spectral feature space to integrate domain-level structures with point-wise physical states. The author claims that HPM achieves strong performance in scarce-data scenarios, offers resolution generalizability, and maintains computational efficiency. To validate these claims, the author conducts experiments across multiple PDE problems, comparing HPM with existing spectral and attention-based neural operators.

**Claims And Evidence:**

The author makes three primary claims: (1) Strong performance and generalizability in data-scarce scenarios, attributed to the spectral priors; (2) Improved training efficiency and the ability to capture fine-scale variations due to point-wise adaptivity; (3) HPM’s learned spectral processing patterns provide insights for designing fixed spectral neural operators. Table 1 and Table 3 compare performance across structured and unstructured meshes, while data scarcity is partially explored in the Darcy Flow and Navier-Stokes experiments. Resolution generalizability is evaluated in zero-shot airfoil cases, though further discussion on generalization under diverse PDE conditions would further strengthen the claim.

**Essential References Not Discussed:**

The author discusses the majority of the essential references.

**Experimental Designs Or Analyses:**

Experimental design has thorough considerations that include structured and unstructured mesh PDEs, generalizability assessments in zero-shot resolution, and the impact of training data. In addition, the author considers ablation studies in spectral coupling functions. One improvement could be the inclusion of some real-world data, such as climate data or smoke plume data, to further demonstrate the robustness of HPM.

**Methods And Evaluation Criteria:**

The benchmark datasets include nine PDE problems, covering both structured and unstructured meshes. The author ensures a comprehensive comparison by including a wide range of existing spectral and attention-based methods. The evaluation metric used is relative L2 error, which is a standard and appropriate measurement in PDE learning, allowing for fair comparisons across different resolutions and problem settings.

**Other Comments Or Suggestions:**

Comments and suggestions are addressed in the earlier sections.

**Other Strengths And Weaknesses:**

Strengths: The limitations and future work are well-discussed, and the paper flows smoothly, making readers easy to follow.

**Questions For Authors:**

Question 1: Beyond resolution generalization, would the design of HPM still be beneficial in other generalization scenarios, such as PDEs with different physical parameter settings?

**Relation To Broader Scientific Literature:**

The broader scientific literature relevant to HPM includes spectral-based and attention-based PDE learning methods. HPM contributes a new perspective by bridging these two categories, integrating spectral priors with point-wise adaptivity to enhance performance.

**Theoretical Claims:**

The theoretical claims in the paper are concisely formulated, mainly presenting short representations of key properties rather than extensive derivations. Specifically, the theoretical claims related to the coupling functions are presented in Equations 11-15, defining how spectral features interact with point-wise physical states.

---

> ### Author Rebuttal · Authors · 2025-04-01
>
> Thank you for your thoughtful review and insightful questions. They have significantly improved this work. We respond to them below.
>
> **#1 Discussion about other generalization scenarios**
> > Question 1: Beyond resolution generalization, would the design of HPM still be beneficial in other generalization scenarios, such as PDEs with different physical parameter settings?
>
> Thank you for this insightful question. Below we provide further discussion about the generalization capability of HPM.
>
> (a) First, we wish to clarify that the strong resolution generalization and limited-data performance naturally stem from the preservation of spectral bias. As derived in Section 3.1, HPM inherits the established inductive bias of fixed neural operators (like FNO) for learning continuous operator mappings. Beyond this, the coupling mechanism allows adaptive spectral modulation based on local physical features, improving the flexibility while maintaining such inductive bias. While not specifically optimized for other generalization scenarios, this unified approach creates fundamental advantages applicable to various generalization tasks.
>
> (b) For specific generalization scenarios, as a versatile neural module, HPM could potentially be combined with dedicated techniques (such as hypernetworks or meta-learning approaches [1,2]) to enhance generalization across different physical parameters. This integration would be a promising direction for future, leveraging HPM's unified representation alongside specialized parameter generalization methods.
>
> **#2 More real-world scenarios beyond standard benchmarks**
> > Experimental design has thorough considerations that include structured and unstructured mesh PDEs, generalizability assessments in zero-shot resolution, and the impact of training data. In addition, the author considers ablation studies in spectral coupling functions. One improvement could be the inclusion of some real-world data, such as climate data or smoke plume data, to further demonstrate the robustness of HPM.
>
> Thanks for your recognition for our comprehensive experiments.
>
> (a) First, We'd like to highlight that the standard benchmarks encompass diverse physical scenarios and have been widely adopted by the community [4,5]. They can effectively validate HPM's capabilities across various physical domains, from fluid dynamics to elasticity problems, demonstrating its general applicability and superior performance compared to existing methods.
>
> (b) Additionally, our current experiments already include problems derived from real-world industry scenarios. For example: (1) The Composite problem simulates deformation fields of Carbon Fiber Reinforced Polymer under high-temperature conditions, directly relevant to aerospace manufacturing of jet air intake components. (2) The Blood Flow problem models hemodynamics in the human thoracic aorta, which has significant clinical applications. These real-world examples demonstrate HPM's practical utility beyond synthetic benchmarks.
>
> We agree that expanding to additional domains like climate data or smoke plume simulations would further validate HPM's robustness. We plan to explore more real-world applications in future work to further demonstrate the practical benefits of our unified spectral-physical approach.
>
> **Reference**
> - [1] Meta-Auto-Decoder for Solving Parametric Partial Differential Equations
> - [2] HyperDeepONet: learning operator with complex target function space using the limited resources via hypernetwork
> - [3] Learning Neural Operators on Riemannian Manifolds
> - [4] Fourier Neural Operator for Parametric Partial Differential Equations
> - [5] Transolver: A Fast Transformer Solver for PDEs on General Geometries

---

### Official Review · Reviewer_13wp · 2025-03-14

**Overall Recommendation:** 4

**Summary:**

The paper introduces Holistic Physics Mixer (HPM), a framework that integrates spectral transformation and data-dependent modulation (i.e. attention). HPM employs a learnable coupling mechanism that enables adaptive modulation of spectral components while preserving the advantages of spectral transformation. The numerical benchmark demonstrates the proposed model outperforms other baselines on both uniform grid and unstructured grid.

**Claims And Evidence:**

The majority of the claims are numerically tested and validated.

**Essential References Not Discussed:**

Despite the author cited AFNO in the paper, I think it is worth more in-depth discussion and if possible, adding a empirical comparison, since AFNO is also a model combing spectral transformation and data-dependent modulation.

**Experimental Designs Or Analyses:**

While the experiments provided cover quite a lot baseline and ablation studies are pretty comprehensive, the following aspects are not entirely clear to me.

1.	What is the computational cost of precomputing the LBO? And how is the number of bases being chosen?
2.	As HPM can be viewed as an extension of linear attention, its computational complexity should also be similar to linear attention based model like Galerkin transformer/OFormer/GNot. Yet, it is shown in the paper that it is faster than Transolver (which computes attention on only a few slices) and only a small overhead over SpecSolver which does not have any linear attention part in it.

**Methods And Evaluation Criteria:**

The paper benchmarked on benchmark problems that are widely adopted by the community, the selected baseline models are also comprehensive.

**Other Comments Or Suggestions:**

See Experimental Designs Or Analyses.

**Other Strengths And Weaknesses:**

I think the overall idea is straightforward but interesting and definitely a meaningful addition to the community. The strong performance of SpecSolver itself also hints the potential of this direction.

**Questions For Authors:**

See Experimental Designs Or Analyses.

**Relation To Broader Scientific Literature:**

The key contribution can be extended to other applications that involves non-Euclidean geometries like sphere, or tasks beyond PDE modelling such as image modelling.

**Theoretical Claims:**

The theoretical result is based on the universal approximation theorem of neural operator in Kovachki et al., 2023.

---

> ### Author Rebuttal · Authors · 2025-04-01
>
> Thank you for your thoughtful review and insightful questions. They have significantly improved this work. We respond to them below.
>
> **#1 Computational cost of LBO**
> > What is the computational cost of precomputing the LBO?
>
> The LBO eigenfunctions is computed efficiently using the robust-laplacian library, with `near-linear time scaling relative to mesh nodes` for both structured and unstructured meshes. Our empirical evaluation on a single Intel Xeon CPU demonstrates the favorable scaling:
> |Nodes Number|Computation Time (s)|
> |---|---|
> |2.5K|$0.008\pm 6.0\times10^{-6}$|
> |10K|$0.037\pm 1.3\times10^{-4}$|
> |40K|$0.177\pm 4.3\times10^{-4}$|
> |160K|$0.894\pm 4.4\times10^{-3}$|
> |640K|$4.581\pm 9.4\times10^{-2}$|
> |2.56M|$19.86\pm 9.0\times10^{-2}$|
>
> Importantly, this computation is a one-time cost per physical domain, with eigenfunctions reusable across simulations in that domain. It is negligible cost for applications with fixed geometry but varying ICs.
>
> Future work could explore more methods (e.g., learning-based approaches, and FFT-like approaches for uniform grids) to further enhance the computational efficiency.
>
> **#2 Chosen of frequency number**
> > And how is the number of bases being chosen?
>
> The number of frequency bases $k$ in HPM is chosen based on several considerations to balance representational capacity and fair comparison.
>
> For most structured mesh problems, we use $k=128$, while unstructured mesh problems use $k=64$. This aligns with: (a) We align $k$ with the hidden dimension to ensure fair comparison with linear attentions, where $k$ corresponds to $Q$, $K$ dimensions. (b) We maintain consistency with spectral methods [1,2], which typically use comparable total frequency numbers. (c) Following [2,3], we make problem-specific adjustments - set $k=128$ for Navier-Stokes despite its larger hidden dimension (256) to control parameter count, and set $k=32$ for Blood Flow due to limited mesh nodes.
>
> Table 9 confirms HPM performs well across different $k$ values, demonstrating the effectiveness of HPM regardless of basis count. While more sophisticated basis selection strategies (such as AFNO [4]) could be explored in future work, current results effectively demonstrates the benefits of spectral-physical integration.
>
> **#3 Computational complexity comparison with linear attention**
> > As HPM can be viewed as an extension of linear attention, its computational complexity should also be similar to linear attention ...
>
> Below we provide a concise analysis comparing HPM with linear attentions.
>
> (a) HPM and linear attention share similar computational complexity. Linear attention operates at $O(Nd^2)$ for features $\mathbf{x} \in \mathbb{R}^{N \times d}$, while HPM requires $O(Ndk)$ operations, where $k$ represents number of frequency basis. The main computations include computing coupling function at $O(Nk)$ cost, and forward/inverse transforms at $O(Ndk)$.
>
> Since $k$ is typically much smaller than $N$ and comparable to $d$ (e.g., $k=d=128$, $N=11271$ for Airfoil), HPM maintains efficiency like linear attentions.
>
> (b) Our measurements below confirm this analysis. On Airfoil, Galerkin attention requires 16.5ms for inference compared to HPM's 17.4ms. This minor overhead enables HPM to capture both domain-level structure and point-wise adaptivity, delivering significant advantages.
> |Model|Inference Time|Training Time / Epoch|
> |---|---|---|
> |Galerkin Attention|16.5 ms|45.25 s|
> |HPM|17.4 ms|46.11 s|
>
> **#4 In-depth discussion about AFNO**
> > Despite the author cited AFNO in the paper, I think it is worth more ...
>
> Thanks for this insightful suggestion. We'll include a comprehensive discussion about AFNO in the methodology section of final version.
>
> While AFNO [4] also incorporates data-dependent modulation, HPM differs in several aspects: (a) Objective: AFNO aims to improve "efficiency and robustness" of spectral features by sparsifying the frequency modes. In contrast, HPM focuses on "enhancing the flexibility of preset spectral features" via point-wise modulation. (b) Methodology: AFNO selects and modifies modes in the spectral domain post-transformation, while HPM introduces spatial domain modulation pre-transformation, enabling point-wise flexibility while preserving spectral structure. (c) Application: AFNO targets computer vision tasks, whereas HPM addresses PDE challenges requiring both global spectral coherence and local adaptivity.
>
> The spectral processing technique from AFNO could complement HPM in future work, potentially combining both benefits (AFNO's efficiency, HPM's flexibility) for diverse applications.
>
> **Reference**
> - [1] Fourier Neural Operator for Parametric Partial Differential Equations
> - [2] Learning Neural Operators on Riemannian Manifolds
> - [3] Transolver: A Fast Transformer Solver for PDEs on General Geometries
> - [4] Adaptive Fourier Neural Operators: Efficient Token Mixers for Transformers

---

### Decision · Program_Chairs · 2025-05-01

**Decision:**

Accept (poster)

**Comment:**

The paper introduces a framework to integrate spectral and local information within a unified space for neural operators (in emulating systems such as PDEs). Their framework enforces domain-level priors (through spectral transforms) and point-wise flexibility that neither spectral-only models nor attention-only models offer.

Strengths identified:
* Simple idea that shows good performance advantages on many PDE benchmarks (uniform and non-uniform grids) and can be extended to tasks beyond PDE modeling (images) and problems with non-euclidean geometries
* Well written paper with comprehensive evaluations/validations on benchmark PDEs

Weaknesses:
* The main weakness was limited evaluations on real-world scenarios.
* While the idea is simple/novel (which is a key strength), some performance advantages might be minimal owing to simpler benchmarks (2D PDEs etc). However, the diversity of PDE tasks looked at was identified as a strength. Other non-PDE domains can also benefit and the paper could have benefitted from benchmarks in the imaging community. These limitations are acknowledged well in the paper.

Overall, the reviewers found sufficient merit to converge to acceptance of the paper. Some concerns/clarifications on computational cost were raised and answered during the rebuttal.